



# Spectral performance analysis of the Fizeau interferometer onboard ESA's Aeolus wind lidar satellite

Michael Vaughan[1], Kevin Ridley[2], Benjamin Witschas[3], Oliver Lux[3], Ines Nikolaus[4], and
Oliver Reitebuch[3]

[1]Optical & Lidar Associates OLA, Buckinghamshire, United Kingdom
[2]University of Birmingham, Department of Physics and Astronomy, United Kingdom
[3]Deutsches Zentrum für Luft- und Raumfahrt e.V. (DLR), Institut für Physik der Atmosphäre, 82234 Oberpfaffenhofen,
Germany
[4]University of Applied Sciences Munich, Germany

**Correspondence:** Benjamin Witschas (Benjamin.Witschas@dlr.de)

**Abstract.** This paper presents an extensive investigation of the signal fringe profile for the Fizeau interferometer used in the first spaceborne wind lidar Aeolus, and considers the fundamental implications for the wind measurement accuracy in Aeolus and future systems. The early Aeolus design phase considered that the basic fringe would be made up of a Fizeau instrumental component of $\approx 100$ MHz (FWHM), folded with the laser pulse spectral width of $\approx 50$ MHz (FWHM), both of Lorentzian form. Fringe anomalies observed before the mission and related to surface defects in the interferometer plates, triggered the development of wave-optic methods for analysis of the fringe formation. These methods, herein described in an instructional Appendix, were subsequently found to be essential for rigorous modelling of complex fringes for different physical and optical arrangements. Initial signal returns from Aeolus suggested that the Fizeau fringe profile was in fact broadened with a large Gaussian component. The laser pulse was subsequently shown to have a profile close to Gaussian of $\approx 45$ MHz (FWHM) and thus provided a partial contribution. However, detailed examination of experimental Aeolus fringes constructed from ground return signals, showed a large Gaussian component up to $\approx 130$ MHz (FWHM). Wave-optic modelling established that Fizeau "aperture broadening", of this form and magnitude, would be generated for the input signal beam of 500 $\mu$rad field of view (FOV) set at large angles of incidence (AOI) of 300 $\mu$rad. These findings have strong implications for fringe shift and wind measurement accuracy, as given in the quantum limited Cramér-Rao expression and the paramount importance of minimising line width. Extensive modelling and simulation for the broadened profiles calculated above, shows good agreement with measured Aeolus global wind measurement accuracies, and indicates that loss of signal could be due to beam clipping at the field stop for such large AOI. It is established that optimisation of the present Aeolus Fizeau parameters could lead to a factor of 2.5 improvement in wind measurement precision. Future upgrades of the Fizeau interferometer and the laser within reasonable parameters, suggest the potential for an factor of 7.6 improvement on the present performance.





## 1 Introduction

On 22 August 2018, the European Space Agency (ESA) launched the first-ever spaceborne Doppler wind lidar, Aeolus, into a sun-synchronous orbit at about 320 km altitude with an orbit repeat cycle of seven days (ESA, 2008; Schillinger et al., 2003). Aeolus carried the Atmospheric Laser Doppler Instrument (ALADIN) as a single payload and operated successfully until April 2023 while additional instrument tests were performed until the completion of the mission in July 2023. ALADIN provided global profiles of the wind component along the instrument's line-of-sight (LOS) direction from the ground up to about 30 km altitude (ESA, 1999; Stoffelen et al., 2005; Reitebuch, 2012; Kanitz et al., 2019; Reitebuch et al., 2020; Straume et al., 2020), mainly aiming to improve numerical weather prediction (NWP) and medium-range weather forecasts (Weissmann and Cardinali, 2007; Tan et al., 2007; Marseille et al., 2008; Horányi et al., 2015; Rennie et al., 2021). Especially wind profiles acquired over the southern hemisphere, the tropics and the oceans contribute to closing gaps in the availability of global wind data which represented a major deficiency in the global observing system before the launch of Aeolus (Baker et al., 2014).

For the use of Aeolus observations in NWP models, a detailed characterization of the data quality as well as the minimization of systematic errors is crucial. Thus, several scientific and technical studies have been performed and published in the meantime, addressing the performance of ALADIN and the quality of the Aeolus data products. In particular, NWP model data (Rennie et al., 2021), airborne wind lidar measurements (Lux et al., 2020; Witschas et al., 2020; Lux et al., 2022; Witschas et al., 2022a), radiosonde observations (Martin et al., 2021; Baars et al., 2020; Borne, M. and Knippertz, P. and Weissmann, M. and Witschas, B. and Flamant, C. and Rios-Berrios, R. and Veals, P., 2024), and various different ground-based instruments have been used to characterize the quality of Aeolus horizontal LOS winds for different periods, different geolocations, and different data products. In addition to that, the ALADIN instrument performance in space was characterized by investigating the laser frequency stability (Lux et al., 2021), the spectral performance of the Fabry-Perot interferometers used to measure wind from the light backscattered from molecules (Witschas et al., 2022b), as well as the performance of the used detectors (Weiler et al., 2021; Lux et al., 2024).

In this paper, we concentrate on the ALADIN Fizeau spectrometer channel that is used to measure wind from atmospheric Mie scattering, which mainly originates from aerosols and clouds leading to narrow-band backscattering signals. Particularly in the last two years, significant advances have been made in the detailed understanding of the spectral performance of the Fizeau instrument, and the many factors that contribute to its resultant spectral line shape, shift, and width. This understanding has enabled the recent development of two analytic algorithms based on a pseudo-Voigt fitting method, and the high-speed four-channel intensity ratio technique $R_4$, both discussed in Witschas et al. (2023b). These algorithms provide significant advances in both statistical accuracy and valid data gathering compared with the currently available techniques originally developed before launch. Additionally, at a more fundamental level, this detailed understanding permits critical evaluation and review of the many design and experimental parameters of the Fizeau interferometer itself and the overall system.

The paper is thus structured as follows: The basic optical architecture of the Aeolus spectrometers is outlined in Sect. 2, with a summary of the Fizeau design parameters. Significant anomalies of the Fizeau interference pattern, the so-called fringe, found in early tests on ground, could not be explained by classical ray optics and could only be replicated by rigorous wave-





optic modelling. Such wave-optic analysis has not previously been applied to Fizeau interferometry and permits rigorous
investigation of all aspects of its optical science and performance. For the benefit of readers, this material is presented in a
semi-tutorial Appendix with programmatic guidance. Sect. 3 then describes successive studies of the experimental line shape
of Aeolus atmospheric and ground return signals. These proved to be notably different from the simple Lorentzian profiles
supposed in the original design and development studies. The observed profiles are well explained by wave-optic modelling,
presented in Sect. 4, with detailed consideration of the illumination conditions, including the field of view (FOV) and the angle
of incidence (AOI). These results are important for detailed signal-to-noise (SNR) and quantum-limited statistical accuracy.
These aspects are summarised in the final Sect. 5 together with guidance for future systems of improved performance.

## 2 The Fizeau spectrometer on Aeolus

### 2.1 Instrumental design

The instrumental architecture of ALADIN is sketched in Fig. 1. In this paper, attention is directed to the Fizeau interferometer
and the optical components that can have an impact on its performance. The setup of the rest of the instrument is only touched
upon. A more detailed description of the ALADIN instrument itself is given in ESA (2008) and Reitebuch et al. (2018). The
laser transmitters, and their frequency stability, are discussed by Lux et al. (2020, 2021) and the ALADIN spectral performance
and corresponding instrumental drifts are discussed in Witschas et al. (2022b). ALADIN carried two fully redundant laser trans-

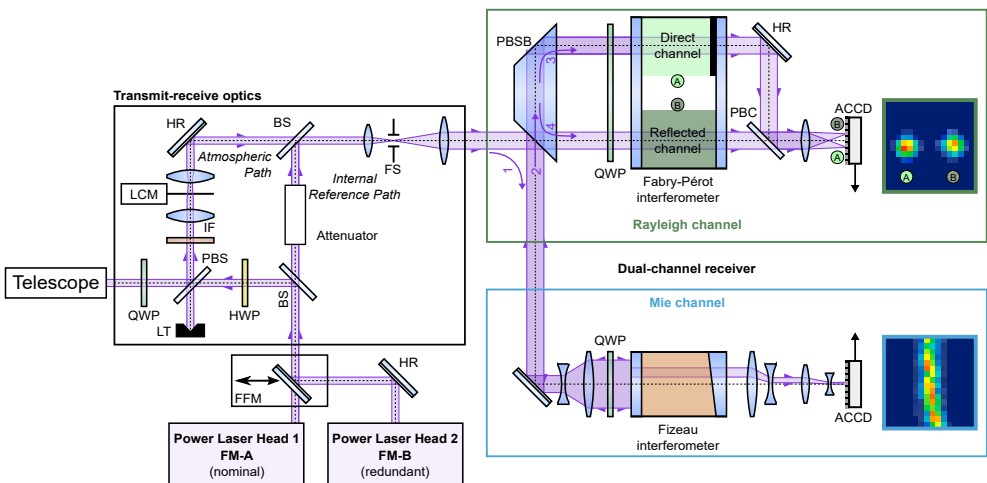

**Figure 1.** Sketch of the ALADIN optical receiver layout reproduced from Lux et al. (2021). QWP: quarter-wave plate; HWP: half-wave
plate; PBS: polarizing beam splitter; PBSB: polarizing beam splitter block; PBC: polarizing beam combiner; FFM: flip-flop mechanism;
BS: beam splitter; HR: high-reflectance mirror; LCM: laser chopper mechanism; FS: field stop; IF: interference filter; LT: light trap; ACCD:
accumulation charge-coupled device.

mitters, referred to as flight models A (FM-A) and B (FM-B), emitting laser pulses at a wavelength of 354.8 nm (vacuum)





and which are switchable by means of a flip-flop mechanism (FFM). After passing through a beam splitter (BS), a half-wave plate (HWP) used to define the polarization of the laser light, a polarizing beam splitter (PBS) used to separate transmitted and received light, and a quarter-wave plate (QWP) setting the transmitted laser light to circular polarization, the laser beam is expanded and coupled out using a 1.5 m diameter Cassegrain telescope. To monitor the frequency of the outgoing laser pulses and to characterize the frequency-dependent transmission functions of the interferometers, a small portion of the laser radia-

tion that leaks through the beam splitter is further attenuated and used as internal reference signal (Fig. 1, Internal Reference Path). The backscattered radiation from the atmosphere and the ground is collected by the same telescope that is used for emission (mono-static configuration) and is returned to the transmit-receive optics (TRO) where a laser chopper mechanism (LCM) is used to protect the detectors from the signal returned during laser pulse emission, after a narrow-band interference filter (IF) with a width of about 1 nm has blocked the broadband solar background light spectrum. Furthermore, the TRO contain a field

stop (FS) with a diameter of 88 $\mu$m to set the FOV of the receiver to be only 18 $\mu$rad which is needed to limit the influence of the solar background radiation and the range of angles incident the spectrometers.

Behind the TRO, the light is directed to the interferometers that are used to analyze the Doppler frequency shift of the backscattered light to finally derive the wind speed along the LOS direction of the laser beam. The light is first directed to the so-called Mie channel via a polarizing beam splitter block (PBSB). After increasing its diameter from 20 mm to 36 mm using a

beam expander, which reduces its divergence from 1 mrad to 555 $\mu$rad, the light is directed to the Fizeau interferometer which acts as a narrow-band filter with a full width at half maximum (FWHM) of 58 fm (138 MHz) to analyze the frequency shift of the narrow-band Mie backscatter from aerosol and cloud particles. The Fizeau interferometer spacer is made of Zerodur to benefit from its low thermal expansion coefficient. It is composed of two reflecting plates, separated by 68.5 mm, corresponding to a free spectral range (FSR) of 0.92 fm (2191 MHz), which is chosen to be a fifth of the FSR of the Fabry-Perot interfer-

ometers (FPIs) used in the Rayleigh channel. The plates are tilted by 4.77 $\mu$rad against each other and the space in between is evacuated. The resultant interference patterns (fringes) are imaged onto the image zone of an accumulation charge-coupled device (ACCD) detector which has 16 x 16 pixels Weiler et al. (2021); Lux et al. (2024). Different laser frequencies interfere at different lateral positions along the tilted plates, so that the horizontal position of the fringe on the detector is a measure of the frequency of the light incident on the Fizeau. The ACCD does not image the entire spectral range covered by the aperture but

only a part of 0.69 fm (1.6 GHz) which is called useful spectral range (USR). This so-called fringe imaging technique using a Fizeau interferometer (McKay, 2002) was specially developed for ALADIN (ESA, 1999).

The accumulated detector signal is converted into a voltage at the ACCD output and afterwards amplified and digitized. Before digitization, an electronic offset voltage - the so-called detection chain offset (DCO) - is applied to prevent negative values in the signal (Lux et al., 2024).

The light reflected from the Fizeau interferometer is directed towards the so-called Rayleigh channel on the same beam path and linearly polarized in such a direction that the beam is now transmitted through the PBSB. The Rayleigh channel is based on the double-edge technique (Chanin et al., 1989; Flesia and Korb, 1999; Gentry et al., 2000), where the transmission functions of two FPIs are spectrally placed at the points of the steepest slope on either side of the broadband Rayleigh-Brillouin spectrum originating from molecular backscattered light. Further details on the FPI specifications on operation principle are





given in Witschas et al. (2022b). For the sake of completeness, the main specifications of the Fizeau interferometer are listed in table 1.

**Table 1.** Specifications of the Mie spectrometer of the ALADIN instrument.

| Parameter | Value |
|---|---|
| Material | Zerodur |
| Aperture | 36 mm |
| Plate spacing | 68.5 mm, vacuum gap |
| Free spectral range | 0.92 pm, 2191 MHz |
| Wedge angle | 4.77 $\mu$rad |
| Plate reflectivity (in air) | 0.85 |
| Plate reflectivity (vacuum) | 0.88 |
| Useful spectral range | 0.69 pm, 1.6 GHz [a] |
| Fringe FWHM | 0.0575 pm, 137 MHz [a] |
| Input divergence | 555 $\mu$rad full angle [a] |

[a] Value taken from Reitebuch et al. (2009)

## 2.2 Initial findings for the Aeolus Fizeau interferometer from on-ground characterisation

In the Fizeau interferometer, light is successively reflected between the surface coatings of the two plates set at the required wedge angle. Multiple interference occurs, ideally leading to straight-line fringes parallel to the wedge vertex. Unlike the FPI,
these fringes are localized close to the plate surfaces and are often described as fringes of equal thickness. In the ray optic approximation the fringes may be considered to trace out the loci of constant path separation between the plates - thus, giving straight line fringes for ideally flat plates. In practice, plates are not perfectly flat, however, minor defects of order $\lambda/100$ across the plates are usually considered as adding to the fringe width in a relatively minor and acceptable degree. Detailed analysis of Fizeau fringes has long been carried out by techniques of ray optics as given for example in the classical text of Born and
Wolf (1980) drawing on the analysis of Brossel (1947) and developed by many subsequent authors (Meyer, 1981; Kajava et al., 1993; McKay, 2002).

The plates selected for the ALADIN spectrometer were polished in the early 2000s, by the relatively new technique of Magneto-Rheological Finishing (MRF) (Jacobs et al., 1995; Harris, 2011); the impact of this polishing technique on the Fizeau interferometer performance was extensively investigated by Vaughan and Ridley (2013); Vaughan and Ridley (2016). In the
MRF technique, the surface is polished by tracing over the optical element with a comparatively small region of magnetically stiffened cutting medium. For the circular Fizeau plates, the cutting medium was traced in a spiral pattern across the surface. Using the MRF technique, a surface finish/roughness of less than 1 nm is expected. Optical examination and tests confirmed that the overall flatness and smoothness of the plates fell within the specification of better than $\lambda/100$, equivalent to $\approx 4$ nm. However, a detailed interferometric examination showed clear evidence of a regular character to the defects with a circular, ring-





like structure. These successive rings/spirals appear to be centered approximately at the centre of the plates. Initial estimates suggested that the pitch, which describes the radial distance between the rings, was $\approx 1 \, \mathrm{mm}$ with a depth of $\approx \pm 4 \, \mathrm{nm}$. This $1 \, \mathrm{mm}$ pitch is consistent with the cutting interval of the MRF polishing technique as it spirals over the plate (Vaughan and Ridley, 2016). In contrast, classical polishing techniques are different. Here, defects of $\lambda/100$ might be expected, but spread in a single cycle across the full area of the plates to give a weak departure from flatness - often described as "dishing" or

"bowing". In the MRF technique, however, the plate surface is much more rapidly corrugated with a peak-to-valley distance for the defects of order $0.5 \, \mathrm{mm}$, which corresponds to half of the pitch.

Initial examination in the laboratory of the Fizeau fringes revealed two rather unusual findings (Francou et al., 2017). First, the fringes, rather than being generally uniform and approximately straight lines, were strongly modulated and appeared to be broken up along their length into regions of high and low intensity. Second, as the input frequency was varied so that the

fringe moved laterally across the plates, these regions of high and low intensity traced out what appeared to be equi-spaced circular rings with a center close to the center of the plates. It thus became imperative to examine the potential impact of these findings on the spectroscopic performance of the Fizeau interferometer. The immediate concern was the potential distortion of the vertically integrated fringe profiles and resultant frequency shifts, which could lead to significant errors in the frequency measurement and the wind velocity accuracy. It was rapidly established that classical techniques of ray optic analysis, which

do not account for diffraction and changes of local slope at the plate defects, could not explain the observed fringe anomalies. Accordingly, a novel wave-optic technique (see e.g. Jakeman and Ridley, 2006) was introduced and shown to accurately reproduce the observed fringes. The development of these methods and extension to the optical science and performance of Fizeau interferometers is detailed in the Appendix and provides an underlying framework for the following sections.

## 3 Examination and analysis of Aeolus Fizeau fringes

From the most basic consideration of the Aeolus Fizeau interferometer, the form of the raw signal fringe profile $\mathcal{P}_{\mathrm{Raw}}$, as it emerges from the detector, may be derived from

$$\mathcal{P}_{\mathrm{Raw}} = \mathcal{P}_{\mathrm{Las}} * \mathcal{P}_{\mathrm{Fiz}} * \mathcal{P}_{\mathrm{Det}}, \qquad (1)$$

where $*$ denotes the convolution by the folding integral, $\mathcal{P}_{\mathrm{Las}}$ the laser pulse profile, $\mathcal{P}_{\mathrm{Fiz}}$ the Fizeau instrument profile, and $\mathcal{P}_{\mathrm{Det}}$ the detector channel spectral profile. Eq. (1) gives a continuous profile. The actual discrete detector outputs can be found

by evaluating it at locations corresponding to the centres of the detector pixels. Note, also, that if the frequency of the input laser light is varied in small steps, values of $\mathcal{P}_{\mathrm{Raw}}$ can be found at sub pixel intervals (see e.g. Marksteiner et al., 2018, 2023).

The ALADIN laser transmitters were developed and built by the company Selex Galileo (today Leonardo) who characterized $\mathcal{P}_{\mathrm{Las}}$ by early laboratory measurements to be smaller than $50 \, \mathrm{MHz}$ (FWHM) with a supposed Lorentzian spectral shape (Cosentino et al., 2012, 2017). The Fizeau interferometer was manufactured by the company Thales-SESO (Francou

et al., 2017). Its design specifications, notably plate reflectivity and wedge angle, were selected to minimize the inherent asymmetry of the Fizeau fringes [see also Section 2]. It was thus considered that the basic instrumental profile for monochromatic





input would be close to the Airy form [see also Eq. (8)], which can be conveniently written as a sum of successive Lorentzians, spaced by the free spectral range $\Gamma_{\mathrm{FSR}}$. With the specification of $\Gamma_{\mathrm{FSR}} = 2191$ MHz and a plate reflectivity $R = 0.88$ (in vacuum), the equivalent single Lorentzian profile representing $\mathcal{P}_{\mathrm{Fiz}}$ would have a width of $\Gamma_{\mathrm{FSR}} \cdot (1 - R)/(\pi\,R^{1/2}) \approx 90$ MHz.
Further, with the selected fringe imaging lens, the 16 detector channels closely approximate a rectangular "top-hat" function $\mathcal{P}_{\mathrm{Det}}$ with a width of $\approx 100$ MHz. On this analysis, with $\mathcal{P}_{\mathrm{Las}}$ and $\mathcal{P}_{\mathrm{Fiz}}$ both having a Lorentzian spectral shape, their combined profile would also be Lorentzian with a width given by their linear sum equal to $\approx 140$ MHz. The folding integral of a Lorentzian and a top-hat function has a width given by the root sum of squares, which leads to a resultant FWHM for the full raw profile $\mathcal{P}_{\mathrm{Raw}}$ of $\approx 172$ MHz. These considerations have provided the basis for the original analytical algorithm which was
developed for the analysis of Aeolus fringes, and which has been refined through successive improvements and upgrades (Reitebuch et al., 2018). In essence, it applies a best-fitting procedure of a pixelated Lorentzian to the measured fringes after the signal has been corrected for the DCO and the solar background signal.

In summary, the foregoing parameters immediately indicate the problems of reliable, unbiased analysis. The actual observed channel contents from the detector are highly averaged representations (resolution of $\approx 100$ MHz) of the incident fringe
profile (width of $\approx 140$ MHz). Inevitably, any fine detail is irretrievably lost, and any analytic technique for derivation of frequency shift and width will have some bias and inaccuracies depending on the assumed model of the fringe profile and how closely representative it is of the true fringe. These errors are likely to be reduced for model profiles that most accurately match the actual profile. This provides an additional underlying rationale for the present investigations.

### 3.1 Further contribution to the Fizeau fringe profile

Several other factors can make a greater or lesser contribution to the Fizeau interferometer output profile $\mathcal{P}_{\mathrm{Raw}}$ and need to be considered. A partial listing would include, for example, the spectral character of the incoming light field, such as spurious background and laser frequency instability and jitter. Other important considerations are the physical characteristics of the incoming beam including AOI, FOV, and speckle effects; the non-uniform illumination of the Fizeau plates; and the impact of plate defects. Additionally, residual asymmetries in the interferometric Fizeau profile may require correction, while
detector performance issues—such as pixel width non-uniformity, quantum efficiency variations, edge effects, spill-over, and charge transfer efficiency-can also impact results. These factors are discussed in greater detail in the following sections where relevant. However, one factor has an overall influence on $\mathcal{P}_{\mathrm{Raw}}$, namely the non-uniformity of plate illumination. Unlike FPI fringes, the Fizeau fringe is localized in the plane of the Fizeau plates which must then be focused onto the detector plane. Thus, the precise form of the Fizeau fringe registered by the detector is strongly impacted by any lack of uniformity of the
incident illumination at the plates. The Aeolus internal reference beam is well established as non-uniform (see e.g. Witschas et al., 2022b) and, without considerable post-detection correction, leads to distorted fringes (Francou et al., 2017).

In comparison, Aeolus ground and atmospheric returns should provide uniform illumination at the entrance to the telescope, but this is of course subject to obscuration within the telescope optics, most notably the secondary mirror and its support structures. Various early analyses based on simple geometrical considerations of the obscuration were attempted but are now
superseded by the more soundly based EMSR (effective Mie spectral response) correction (Wang et al., 2024). In this deriva-





tion, it is considered that the broadband background signal, following a Rayleigh-Brillouin (RB) spectral distribution (Witschas et al., 2010; Witschas, 2011a, b), is close to spectrally uniform (i.e. flat) across the Fizeau spectral range. Hence, by averaging and comparing Fizeau channel contents from areas dominated by pure RB signals, a good characterization of the obscuration in the Fizeau telescope optics was derived and used for correction. Subsequent wave-optic modelling of the overlap of orders for the RB spectrum established that the background across one order was indeed completely flat (see also Sect. 4.5). It is worth mentioning, that the EMSR does not only correct for the obscuration, but also for the actual illumination of the Fizeau interferometer and its temporal evolution. The EMSR correction thus provides the possibility of retrieving the Fizeau fringe spectral shape with high accuracy. This is particularly important for strong ground returns which should be essentially monochromatic with no additional Mie or Rayleigh response. Based on this, fringes from ground return signals were acquired during instrument response calibration (IRC) measurements. An IRC is performed with the instrument LOS pointing in nadir direction and changing the laser frequency in steps of 25 MHz over a spectral range of 1000 MHz (Marksteiner et al., 2018, 2023). The resulting ground return signals of such IRC measurements enabled the construction of prototype Fizeau fringes and the detailed analysis of their spectral characteristics as it is discussed in the following sections.

### 3.2 Fizeau fringe characterization by nonlinear fit procedures

To analyze the spectral characteristics of the Aeolus Fizeau fringes in detail, internal reference signals (INT) as well as atmospheric ground return signals (ATM) from an IRC measurement performed on 4 July 2019 are used as shown in Fig. 2. To avoid the influence of broadband RB background in the ATM signal, only the four fringes with the highest signal intensities were chosen and averaged as shown by the black circles in Fig. 2 (b). Panel (a) shows the corresponding fringe from the INT signal, which was also EMSR corrected using the illumination function as it is for instance characterized by instrument spectral registration (ISR) measurements (Witschas et al., 2022b). As the illumination characteristics are different for the ATM and the INT path, different EMSR corrections have to be applied. Furthermore, both signals are corrected for the DCO, and the ATM signal is additionally corrected for the solar background signal. As described above, the Fizeau instrument function as well as the laser profile can approximated by a Lorentzian peak function according to (Born and Wolf, 1980; Vaughan, 1989; Witschas et al., 2023b)

$$\mathcal{L}(x) = \frac{2\mathcal{I}_{\mathcal{L}}}{\pi} \cdot \frac{\Gamma_{\mathcal{L}}}{4(x - x_0)^2 + \Gamma_{\mathcal{L}}^2}, \tag{2}$$

where $\mathcal{I}_{\mathcal{L}}$ denotes the area under the peak, $\Gamma_{\mathcal{L}}$ the FWHM, and $x_0$ the center position. The raw fringe profile, as given by Eq. (1), has additionally been convolved with the detector profile $\mathcal{P}_{\mathrm{Det}}$ which can be described by a top-hat function according to

$$\mathcal{P}_{\mathrm{Det}} = \frac{1}{\Gamma_{\mathrm{TH}}} \cdot H(x) \cdot \left( \frac{1}{4} - \left( \frac{x}{\Gamma_{\mathrm{TH}}} \right)^2 \right), \tag{3}$$

where $H(x)$ is the Heaviside step function, and $\Gamma_{\mathrm{TH}}$ denotes the width of the top-hat. The convolution of Eq. (2) and Eq. (3) can be derived analytically and results in a pixelated Lorentzian $\mathcal{L}_{\mathrm{px}}(x)$ according to

$$\mathcal{L}_{\mathrm{px}}(x) = \frac{\mathcal{I}_{\mathcal{L}_{\mathrm{px}}}}{\pi \cdot \Gamma_{\mathrm{TH}}} \cdot \left( \arctan \left( \frac{-2x_0 + \Gamma_{\mathrm{TH}} + 2x}{\Gamma_{\mathcal{L}}} \right) + \arctan \left( \frac{2x_0 + \Gamma_{\mathrm{TH}} - 2x}{\Gamma_{\mathcal{L}}} \right) \right), \tag{4}$$





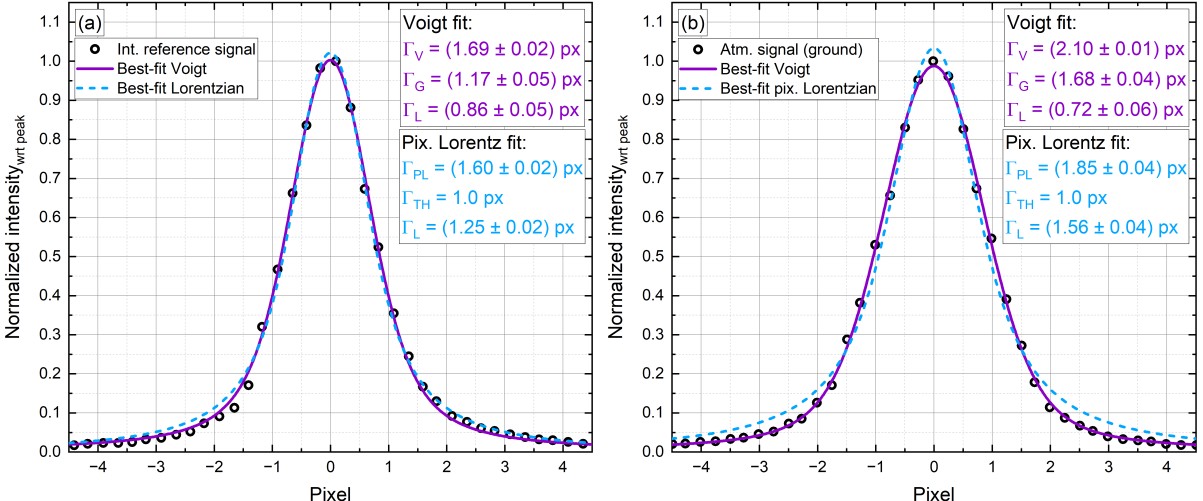

**Figure 2.** Averaged Aeolus Fizeau fringes (EMSR and background signal corrected) depicted by the black circles for the internal reference signal (a) and the ground return signal (b), retrieved from the instrument response calibration (IRC) measurement performed on 4 July 2019. The blue line indicates a best-fit of a pixelated Lorentzian according to Eq. (4) and the purple line indicates a best-fit of a Voigt profile according to Eq. (5). Details of the fit results are given by the inset. See text for explanation of the symbols.

where $\mathcal{I}_{\mathcal{L}_{\mathrm{px}}}$ denotes the area under the peak. Now, Eq. (4) is applied in a least-square fit procedure to the measured prototype fringes as shown by the blue dashed lines in Fig. 2. The resulting fit parameter are given by the inset. It is obvious that the

accordance of the fit with the measured fringe is not good, especially in the wings of the fringe and this is most pronounced for the ATM fringe (Fig. 2 b). The resulting widths are $\approx 160\,\mathrm{MHz}$ for the INT and $\approx 185\,\mathrm{MHz}$ for the ATM signal, which is close to the estimate of $\approx 172\,\mathrm{MHz}$ as given above, when considering a laser pulse profile of width $\approx 50\,\mathrm{MHz}$ and a Fizeau profile of width $\approx 90\,\mathrm{MHz}$, both having a Lorentzian shape. However, the poor accordance of the fit reveals that the actual contributions to the fringe profile are of different nature.

In light of, it was investigated if a Voigt function $\mathcal{V}(x)$, defined as the convolution of a Lorentzian $\mathcal{L}(x)$ (Eq. 2) and a Gaussian peak profile $\mathcal{G}(x)$, represents the prototype fringe with better accuracy:

$$\mathcal{V}(x) = (\mathcal{L} * \mathcal{G})(x), \tag{5}$$

where $*$ denotes the convolution of the folding integral, and

$$\mathcal{G}(x) = \left(\sqrt{\frac{4\ln 2}{\pi}} \cdot \frac{1}{\Gamma_{\mathcal{G}}}\right) \cdot \exp\left(-4\ln 2 \cdot \left(\frac{x - x_0}{\Gamma_{\mathcal{G}}}\right)^2\right), \tag{6}$$

with $\Gamma_{\mathcal{G}}$ being the FWHM.





Although the Voigt function cannot be represented in an analytically closed form, or rather without using special functions, its FWHM $\Gamma_\mathcal{V}$ can be approximated with an accuracy of better than $0.02\%$ according to (Olivero and Longbothum, 1977) by:

$$\Gamma_\mathcal{V} = 0.5346\,\Gamma_\mathcal{L} + \sqrt{0.2166\Gamma_\mathcal{L}{}^2 + \Gamma_\mathcal{G}{}^2}. \tag{7}$$

Equation (5) is used to perform a numerical least-square fit to the prototype fringes as shown by the purple lines in Fig. 2. The
resulting fit parameters are given by the inset. It is obvious that there is excellent accordance between the prototype fringes and the fit for both, the INT and the ATM signals. Both, the slopes and the wing intensity are reproduced very well. The fit yields an FWHM $\Gamma_\mathcal{V}$ of $(1.69 \pm 0.02)\,\mathrm{px}$ or $(169 \pm 2)\,\mathrm{MHz}$ for the INT and $\Gamma_\mathcal{V} = (2.10 \pm 0.01)\,\mathrm{px}$ or $(210 \pm 1)\,\mathrm{MHz}$, for the ATM signal. For the INT signal, $\Gamma_\mathcal{L} = (0.88 \pm 0.05)\,\mathrm{px}$ and $\Gamma_\mathcal{G} = (1.17 \pm 0.05)\,\mathrm{px}$, and for the ATM signal, $\Gamma_\mathcal{L} = (0.72 \pm 0.06)\,\mathrm{px}$ and $\Gamma_\mathcal{G} = (1.68 \pm 0.04)\,\mathrm{px}$. From this, interesting characteristics of the fringes can be derived. First, it can be seen that the
with of the INT fringe (169 MHz) is close to the expectations, however, the ATM fringe is significantly broader (210 MHz). Furthermore, it can be realized that a large Gaussian component has to be considered in order to describe the prototype fringes with sufficient accuracy, which is in contrast to all original expectations. Wave-optic analyses, as later discussed in Sect. 4.2, have revealed, that, for the ATM path, an off-axis illumination of the Fizeau interferometer of $\approx 400\ \mu\mathrm{rad}$ with a divergent laser beam ($\approx 500\ \mu\mathrm{rad}$) can explain the observed Voigt-shape and width of the Aeolus Mie fringes.
The foregoing discussions outline the complexities of Fizeau fringe formation and raise questions about how to usefully resolve them. In order to answer these questions, the underlying components $\mathcal{P}_{\mathrm{Las}}$ and $\mathcal{P}_{\mathrm{Fiz}}$ and the impact of $\mathcal{P}_{\mathrm{Det}}$ are examined in Sect. 3.3, leading to a better understanding of the physical/optical nature and the implications for future design.

### 3.3 Calculation of the basic components of the Aeolus Fizeau interferometer fringe

In the framework of a pre-development programme that was conducted in the early phase of the Aeolus preparation (Du-
rand et al., 2004), laboratory tests of the receiver breadboard were performed, including the characterization of the Fizeau interferometer. These measurements also defined the Fizeau parameters as summarized in Table 1.

Initial laboratory measurements in air suggested experimental line widths of 105 MHz in reasonable agreement with a reflectivity finesse of $N_R = 20.8$ and consistent with a plate reflectivity of $R \approx 0.85$. However, in later measurements in vacuum, line widths somewhat less than 100 MHz were observed. As shown by Stolz et al. (1993), reflectivity changes of
$\approx 3\%$ to shorter wavelengths can appear when going from air to vacuum, due to changes in the dielectric coating layers. Reflectivity versus wavelength curves for a pair of plates were available and showed that the reflectivity in vacuum, for such a $3\%$ wavelength shift, was closer to $0.88$. This latter value has accordingly been used as a good representative value for further investigations discussed in this study.

Furthermore, the Fizeau plates received a detailed examination of surface characteristics revealing not only the semi-regular
fine-scale defects due to MRF finishing but also structures across a larger scale. Wave-optic modelling for these measured defects showed fluctuations of frequency response across the plate in the range of $-10$ MHz to 2 MHz, compared with the input frequency (Vaughan and Ridley, 2013; Vaughan and Ridley, 2016). The apparent FWHM also varied over $105 \pm 7$ MHz.





Later examination of the fringe profile shapes indicated a Gaussian component that could approach up to 20 MHz induced by the aforementioned plate defects.

Before the mission, the spectral pixel width was characterized in different laboratory tests to be in the range of 95 MHz to 105 MHz. Some of this variation was attributed to uncertainties in the precise optical magnification between the detector plane and the Fizeau instrument. Precise investigations of the Aeolus system based on regular IRC measurements (Marksteiner et al., 2023) support a spectral pixel width of 94 MHz to 95 MHz throughout the entire mission time. For the sake of simplicity and without impacting the drawn conclusions, a spectral pixel width of 100 MHz is used throughout this study.

The uncertainties of the fringe position and the spectral profile of these findings are relatively small, of order 10 MHz or less. As such, they are unlikely to explain the considerably larger magnitude of the ATM prototype fringe FWHM of 2.10 px ($\approx 210$ MHz), as discussed in Sect. 3.2. This would in fact require an input Lorentzian of $\approx 182$ MHz, to be folded with the top-hat of 100 MHz (1 px). And even then, the overall fringe profile cannot be described accordingly as shown in Fig. 2.

The following three subsections describe techniques that attempt to analyze the prototype fringe, and to quantify the profiles and magnitude of the individual contributions (laser, Fizeau, detector). These techniques rely on the evaluation and comparison of the pixel contents across the prototype fringe, namely from the total energy within the fringe (Sect. 3.3.1), from the relative pixel content around the fringe peak (Sect. 3.3.2), and from consideration of the pixel content in the outer region of the fringe (Sect. 3.3.3). It may be noted that, for a large detector function with a width comparable to the one of the input fringe,

distortion of the output fringe is large. Commonly applied ratio techniques, using profile widths at different relative intensities, proved liable to error and unpromising. Hence the preference for examination of channel content as detailed below.

### 3.3.1    Calculation of fringe components from total fringe content

From the prototype fringe as shown in Fig. 2, the total content $\mathcal{I}_{\text{fringe}}$ is numerically determined to be $\mathcal{I}_{\text{fringe}} = 2.505$, where the fringe has a peak normalized at unit intensity ($\mathcal{I}_{\text{peak}} = 1$). The corresponding FWHM is determined by a best-fit of Eq. (5)

to the data (Fig. 2 a, purple line) and using Eq. (7), resulting in $\Gamma_{\mathcal{V}} = 2.10$ px. In order to quantify the respective Lorentzian and Gaussian contribution to the Voigt-shaped profile from these values, the Voigt profile table provided by Tudor Davies and Vaughan (1963) is used. This table characterizes the Voigt profile regarding intensity and width for various Lorentzian/Gaussian ratios. Hence, by using the respective values from the prototype fringe as mentioned above, the Lorentzian and Gaussian contribution can be read from this table. For instance, the total fringe content is expressed as $\mathcal{I}_{\text{fringe}} = p \cdot \mathcal{I}_{\text{peak}} \cdot \Gamma_{\mathcal{V}}$, where $p$ is

a numerical value for any specific Voigt function.

From the parameters retrieved for the the prototype fringe, the value of $p = 2.505/(1 \cdot 2.10) = 1.19$. Referring to the Voigt tables, the corresponding fractional values of the components can be read to be $L_{\text{fraction}} = 0.30$ and $G_{\text{fraction}} = 0.83$. Given a FWHM of 2.10 px, the corresponding components are $L_{\text{FWHM}} = 0.63$ px and $G_{\text{FWHM}} = 1.74$ px. With 1 px $\approx 100$ MHz, this initial estimate of $L_{\text{FWHM}} \approx 63$ MHz appears slightly smaller than anticipated, while the $G$-component is somewhat

larger. However, this outcome is reasonable as the procedure essentially "force fits" the prototype fringe using a Voigt profile composed solely of pure $L$ and $G$ components. The pixelated detection introduces the large extra component of a top-hat (1 px





wide). Effectively, this top-hat may be considered as operating as a "super Gaussian" with zero wings. When folded with other functions, notably $L$ and $G$, the top-hat serves to reduce the apparent $L$ component, and enlarge the $G$ component, in the subsequent force fit to a pure Voigt function. It is in fact possible to introduce first order corrections to the above calculation, taking account of the relative changes of peak height and width due to folding with a top-hat function. With these corrections the components are given by $L_{\mathrm{FWHM}} \approx 0.85$ px and $G_{\mathrm{FWHM}} \approx 1.47$ px.

In summary, this straightforward procedure provides strong evidence that the fringe output from the Fizeau instrument has a large Gaussian component of about 147 MHz. Most notably the Lorentzian component of about 85 MHz appears to be close to that calculated for the finesse limited line width of the Fizeau interferometer itself ($\approx 90$ MHz).

### 3.3.2 Calculation of components from individual channel contents close to the peak

In a next step, the individual channel contents in the prototype fringe are examined in terms of their fractional content compared with the content of the full fringe (i.e. the 22 channels of a full FSR). From the prototype records, the content of the individual channels are calculated as a fraction of the total content across the complete fringe. This summation requires that the fringe is considered across the full FSR (closely equivalent to 22 channels, corresponding to $\Gamma_{\mathrm{FSR}} = 2191$ MHz) as compared with the 16 channels of the detector (USR). Simple estimations of the content in these six outer channels amounts to $\approx 1.9\%$ of the complete profile. The resultant corrections across the channel contents close to the peak are less than $0.01$ (fractional unit). The fractional single channel contents, as averaged for the two symmetric, nominally equal channels on either side of the centre, are plotted in Fig. 3. As a first comparison, the pixelated Lorentzian input fringe with $\Gamma_{\mathcal{L}} = 1.82$ px, convolved

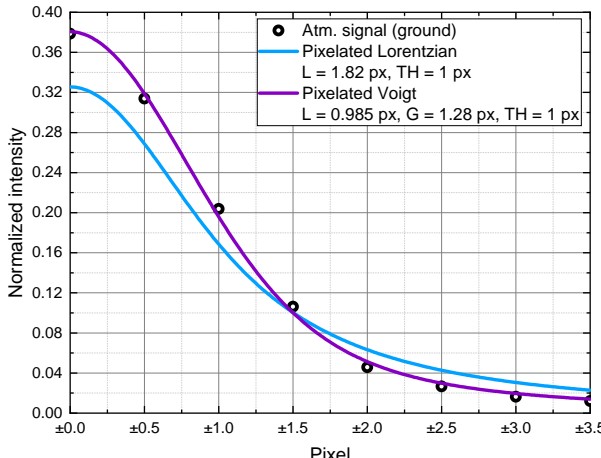

**Figure 3.** Plots of fractional fringe content for the pixels around the peak for the experimental prototype fringe built up from Aeolus ground returns (black circles) - (see also Fig. 2 b), compared with a pixelated Lorentzian according to Eq. (4) with $\Gamma_{\mathcal{L}} = 1.82$ px and $\Gamma_{\mathrm{TH}} = 1$ px (light-blue line) and with a pixelated Voigt function with $\Gamma_{\mathcal{L}} = 0.985$ px and $\Gamma_{\mathcal{G}} = 1.28$ px and $\Gamma_{\mathrm{TH}} = 1$ px (purple line).

with a top-hat function of $\Gamma_{\mathrm{TH}} = 1$ px width is considered (light-blue line) according to Eq. (4), resulting in a total width of $\approx 2.1$ px. For this pixelated Lorentzian, although the width is equal to the one determined for the experimental prototype



fringe (black circles), the calculated fractional contents are obviously different. Most notably, the two central pixels of the prototype fringe, are about $0.05$ (fractional unit) greater. Correspondingly the two outer pixels are more than $0.015$ smaller. The second comparison (purple line) is based on a Voigt function with $\Gamma_{\mathcal{L}} = 0.985$ px and $\Gamma_{\mathcal{G}} = 1.28$ px, giving a FWHM of $\Gamma_{\mathcal{V}} = 1.89$ px [Eq. (7)], numerically folded with a top-hat function of width $\Gamma_{TH} = 1$ px, to result in a total width of $2.1$ px.

The close correspondence of this profile with the prototype fringe values provides further strong confirmatory evidence that the Fizeau fringe before detection is made up of a Lorentzian of about $1$ px ($100$ MHz) and a Gaussian of about $1.3$ px ($130$ MHz).

### 3.3.3 Analysis of the outer part of the fringe

For a detailed analysis of the outer part of the prototype fringe, the Lorentzian formula [Eq. (2)] is no longer a good approximation and the Fizeau fringe is better described by the classical Airy formulation according to (see e.g. Hernandez, 1986;

Vaughan, 1989)

$$I_T = I_0 \cdot \frac{T^2}{(1-R)^2} \cdot \frac{1}{1 + F \sin^2{(\varphi/2)}}, \tag{8}$$

where $F = 4R/(1-R)^2$ is the finesse coefficient, $R$ and $T$ the plate reflectivity and transmission terms, and $\varphi$ the phase lag per optical transit of the plate separation. Note the difference to the commonly used reflectivity finesse which is given by $N_R = (\pi/2) \cdot F^{1/2} = \pi R^{1/2}/(1-R)$. The Airy function $\mathcal{A}(\varphi)$ is accordingly given by $\mathcal{A}(\varphi) = \left[1 + F \sin^2{(\varphi/2)}\right]^{-1}$, with

typical values of $N_R \approx 24.6$ and $F \approx 244$, for a mean plate reflectivity of $R \approx 0.88$. In the outer part of the fringe, i.e. $\varphi/2 \gtrsim \pi/3$, $A(\varphi)$ can be closely approximated by $F^{-1}\left[\sin^2{(\varphi/2)}\right]^{-1}$. As discussed by means of Eq. (1), the detector raw signal is additionally impacted by the laser pulse profile (e.g. a Gaussian) and the detector channel spectral width (e.g. a top-hat). The impact of these contributions to the outer part of the fringe profile is illustrated in Fig. 4, which shows the fringe development of a basic Airy profile of unit height and FWHM of $1$ px (black), convolved with a Gaussian function of width $1.2$ px (magenta)

and further broadened by the top-hat detector function of width $1.0$ px (light-blue). All three curves are normalized to unit area. It can be readily observed, that the convolution of the Gaussian and the top-hat results in negligible changes in the outer part

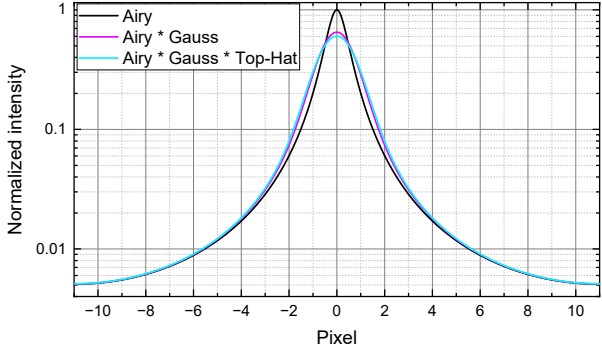

**Figure 4.** Basic Fizeau Airy type fringe (black) convolved with a Gaussian function of width $1.2$ px (magenta) and the top-hat detector function of width $1.0$ px (light-blue). The total energy (i.e. the area) is conserved. The y-axis is in log-scale.





of the fringe. This is due to the fact that the Gaussian and top-hat functions do not have extended wings that would redistribute energy into the outer regions.

In a next step, the Airy profile $\mathcal{A}(\varphi)$ is compared to the outer part of the ATM prototype fringe. From the prototype fringe
data, $F^{-1}$ is determined to be $(4.22 \pm 0.20) \cdot 10^{-3}$, which equates to $F = 237 \pm 11$ and $N_R = 24.2 \pm 0.6$. The corresponding Airy profile (light-blue line) and the ATM prototype fringe data (black circles) are plotted in Fig. 5. The reflectivity finesse

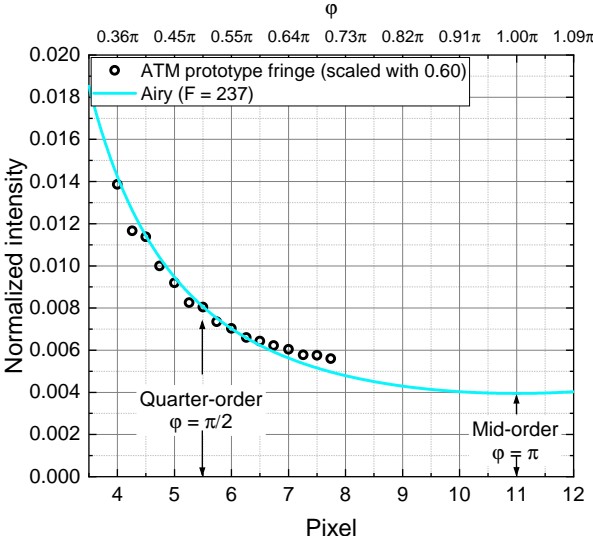

**Figure 5.** Airy fringe according to Eq. (8) over pixels 4 to 11 ($\varphi > 0.35\pi$) for $F = 237$ (light-blue line). The 16 values (black dots) of the ATM prototype fringe (see also Fig. 2 b) were averaged to give a best representative fit for $F$.

$N_R = 24.2$ would suggest that the basic Airy function has a width of $0.91$ px and the associated Gaussian width would be $\approx 1.3$ px to make up the output fringe from the Fizeau, which is then detected as the prototype fringe. It would of course be possible to repeat this evaluation in an iterative procedure with the new starting point of an Airy profile of width $0.91$ px.
However, it does not appear particularly worthwhile. All evidence indicates that the observed fringe initially exhibits a comparatively narrow Airy-type profile, close to Lorentzian form, with a width slightly less than $1$ px ($100$ MHz). This profile is then broadened by successive physical processes that approximate Gaussian and top-hat functional forms.

In summary, the various investigations across the prototype fringe clearly establish that the output fringe from the Fizeau is close to a Voigt function with a large Gaussian component of about $1.3$ px. So, the obvious questions are what are the
physical mechanisms which have led to this unexpected result, and what lessons can be drawn for system performance and improvement?

The outcome of this study further triggered the update of the Aeolus processor, which was still using a pixelated Lorentzian fit to derive the Fizeau fringe positions and the corresponding wind speeds by means of the Mie-core 2 algorithm (Reitebuch et al., 2014). As the Voigt function has no simple analytical solution without special functions, the new Mie-core 3 algorithm





will be based on the pseudo-Voigt approximation, which is a linear combination of $\mathcal{L}(x)$ and $\mathcal{G}(x)$, with identical widths ($\Gamma_{\mathcal{L}} = \Gamma_{\mathcal{G}}$). Based on Aeolus airborne demonstrator data, which has similar characteristics to Aeolus data, Witschas et al. (2023b) demonstrated the much better performance of the pseudo-Voigt fit compared to the Lorentzian fit. In particular, 50% more data points could be reached while keeping the resulting random errors equally sized. In addition, a novel algorithm ($R_4$) has been developed by Witschas et al. (2023b), which is based on a ratio constructed from the four central pixel channels around

the fringe peak. After calibration, the $R_4$ algorithm is demonstrated to provide similar quality as the pseudo-Voigt fit based algorithm, but with a two orders of magnitude faster computation time.

## 4 Modelling and analysis of contributory factors to the Fizeau fringe profile

The previous section has established that the Aeolus Fizeau fringe, prior to detection, is primarily made up of a Lorentzian component of $\approx 100$ MHz ($\approx 1.0$ px) FWHM, folded with a Gaussian component up to $\approx 130$ MHz ($\approx 1.3$ px) FWHM. This

present section investigates the physical/optical basis of these terms and particularly the somewhat unexpected magnitude of the Gaussian component.

### 4.1 The laser pulse profile

The pulse duration $\Delta\tau$ of Aeolus laser pulses was characterized to be $\Delta\tau \approx 20$ ns (Lux et al., 2021; Cosentino et al., 2012). Depending on the actual pulse spectral shape, this corresponds to a Fourier-transform limit of the pulse spectral width (FWHM)

of $\Delta\nu \geq 0.441/\Delta\tau \approx 22$ MHz for a Gaussian-shaped laser pulse, and $\Delta\nu \geq 0.142/\Delta\tau \approx 7.1$ MHz for a Lorentzian-shaped laser pulse (Koechner, 2013). However, heterodyne measurements of the ALADIN Airborne Demonstrator laser transmitter, which is based on a similar configuration with comparable specifications, revealed that the actual line width was approximately twice the Fourier-transform-limit (Schröder et al., 2007). This spectral broadening is attributed to a frequency chirp, most likely caused by changes in population inversion during pulse evolution. As the same effect is assumed for the Aeolus lasers, the

spectral width is expected to be larger than the Fourier-transform limit.

A careful analysis of the intensity spectrum published by Schröder et al. (2007) by width ratio techniques and tables of Voigt integrals, revealed a spectral width of 15.6 MHz (for the infrared beam at 1064 nm), dominated by a large Gaussian component of $(14.7 \pm 0.5)$ MHz, folded with a much smaller Lorentzian component of $(1.7 \pm 0.9)$ MHz. These are derived from the fractional components $\mathcal{G}_{\text{fracction}} = 0.94 \pm 0.3$ and $\mathcal{L}_{\text{fraction}} = 0.11 \pm 0.6$, where the errors are indicative "limit" errors

from the ratios. The measurement of such small Lorentzian fractions is towards the limit of available accuracy.

In conclusion, on frequency tripling from 1064 nm to the operational Aeolus wavelength at 355 nm, one would thus expect the laser pulse profile to be dominated by a Gaussian component of $\approx 45$ MHz FWHM with a Lorentzian component of less than $\approx 5$ MHz.





## 4.2 Wave-optic modelling of FOV and AOI

In the period prior to launch, extensive wave-optic modelling of speckle-type signals and their equivalent optical FOV was carried out (Vaughan and Ridley, 2013; Vaughan and Ridley, 2016). This work largely concentrated on small AOI and questions of apparent frequency shift relative to the input frequency. For single speckle patterns, so-called "frozen speckle", shifts of a few tens of MHz were evident. With appropriate temporal and spatial averaging of speckle, as would be expected for most practical operations, these fringe shifts are reduced by the square root of the number of independent speckle patterns, with 395 small increases in fringe width. These values, evaluated for small AOIs, were thus considered within acceptable bounds.

After launch, evidence steadily accumulated that operational AOIs were indeed considerably larger. For the ALADIN FPIs, AOIs greater than $400\ \mu\mathrm{rad}$ were required to explain the measured fringe widths and shifts, as extensively discussed by Witschas et al. (2022b), particular in their Section 6. This prompted extensive modelling of Fizeau fringes at such larger AOI. Successive steps in this procedure are illustrated in the following diagrams.

Fig. 6 a shows fringe profiles for plane wave illumination with the nominal Aeolus fringe parameters as given in Table 1. At normal incidence (AOI = 0 $\mu\mathrm{rad}$), the resultant fringe is reasonably symmetric (black line). However, at AOI = 300 $\mu\mathrm{rad}$, the fringe is considerably broadened and distorted with a small distinct secondary maximum at the side (purple line). Here, the AOI is defined to be positive when the incoming radiation is tilted towards the apex of the Fizeau wedge. Note that the bottom x-axis is given in millimeter, and the top x-axis indicates the corresponding frequency considering the conversion factor of $59.2\ \mathrm{MHz/mm}$ as used in the wave-optic model. The adjacent Fig. 6 b, for a cone angle illumination FOV = 500 $\mu\mathrm{rad}$,

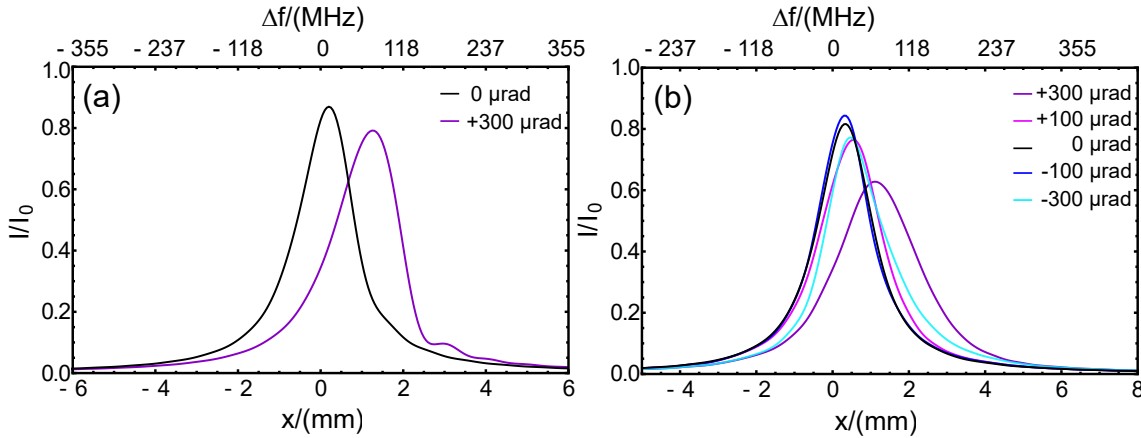

**Figure 6.** Modelled fringe profiles for Aeolus Fizeau nominal parameters as given in Table 1. (a) Plane wave illumination at nominal incidence (black line) and with AOI = 300 $\mu\mathrm{rad}$ (purple line). (b) Illumination with FOV = 500 $\mu\mathrm{rad}$ at different AOI as given by the inset.


used as an approximation of the actual FOV of 555 $\mu\mathrm{rad}$ (see also Table 1), shows model fringes for AOIs up to $\pm 300\ \mu\mathrm{rad}$ in x-tilt, with y-tilt= 0. These have been calculated in a physically realistic way, by starting with an input field consisting of randomly–phased components with a specified angular distribution, i.e. a speckle pattern. Averaging of many uncorrelated fringe intensity patterns mimics temporal integration and produces the final fringe profile. 100 averages are typically sufficient





for a fringe spatially integrated along its vertical axis, i.e. the y–direction. Further details are discussed in the Appendix A3 and A4.

On examination, the fringes for FOV = 500 $\mu$rad (Fig. 6 b) are all reasonably symmetric. Most notably, the secondary maximum shown for the comparable plane wave fringe at AOI = 300 $\mu$rad has been completely smoothed out (compare the purple fringes in Fig. 6 a and Fig. 6 b.). Increased broadening and peak shift for large AOI is evident, and particularly strong for positive AOIs as shown in Fig. 7. Note that the width is FWHM and the shift is calculated as the mid-point of the width, which

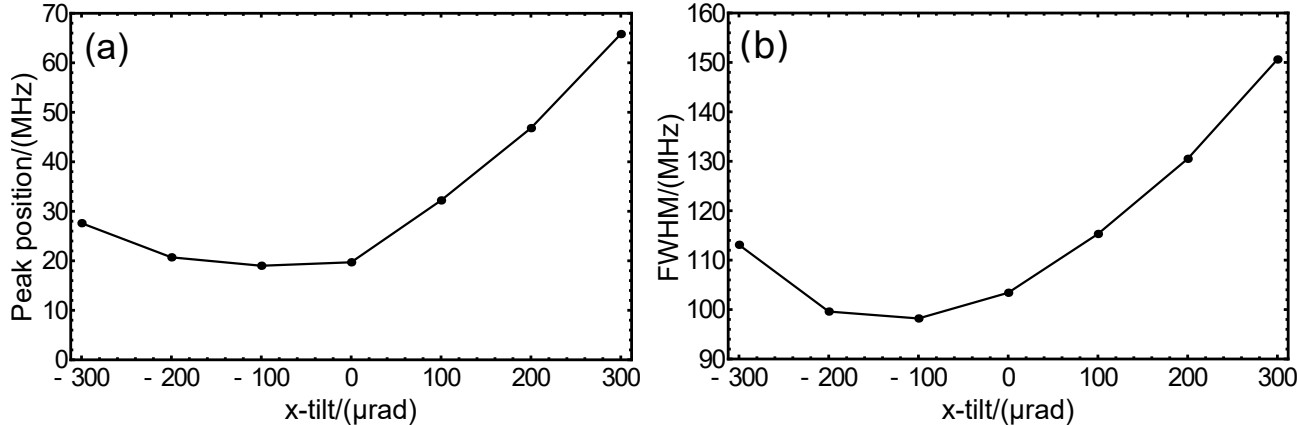

**Figure 7.** Fringe shift (a) and FWHM (b) for the modelled fringes shown in Fig. 6 b. Note the minima for both curves close to AOI = $-100$ $\mu$rad.


provides a good measure of the centre of energy for a fringe having any slight asymmetry. Note also, that the energy within the fringes is essentially constant for different AOI: the calculated changes across the full range are less than 1%. For both, width and shift, the minimum values occur at an AOI close to $-100$ $\mu$rad and not normal incidence. This characteristic has been discussed by Langenbeck (1970) and summarized by McKay (2002), who showed that the optimum angle for illuminating

a Fizeau wedge is tilted away from the apex (negative sign). Equivalent investigations for variation of y–tilt, with x–tilt= 0, showed similar results although, in this case, independent of the sign of tilt angle, with frequency shifts and broadening smaller by a factor of $\approx 0.6$.

Extensive analyses, by ratio techniques and subsequent profile matching, showed that the fringes shown in Fig.6 b are well fitted by Voigt functions. The two examples, for the optimum position with AOI = $-100$ $\mu$rad (x-tilt = y-tilt = 0) and for

AOI = 300 $\mu$rad are shown in Fig. 8 a and Fig. 8 b, respectively, together with fits of a Lorentzian according to Eq. (2) (light-blue line) as well as a Voigt profile according to Eq. (5) (purple line). The respective FWHM derived from the Voigt fit are given by the insets. The better quality of the Voigt fit is particularly notable for AOI = 300 $\mu$rad (Fig. 8 b). It can also be seen that an increase of the AOI increases the overall width by mainly increasing the Gaussian component of the Voigt profile. in particular, for AOI = $-100$ $\mu$rad, the Voigt profile has a width of $\Gamma_\mathcal{V} \approx 98.2$ MHz, being composed of $\Gamma_\mathcal{G} \approx 27.4$ MHz and

$\Gamma_\mathcal{L} \approx 90.0$ MHz. On the other hand, for AOI = 300 $\mu$rad, the Voigt profile has a width of $\Gamma_\mathcal{V} \approx 150.8$ MHz, being composed of $\Gamma_\mathcal{G} \approx 89.7$ MHz and $\Gamma_\mathcal{L} \approx 95.3$ MHz.



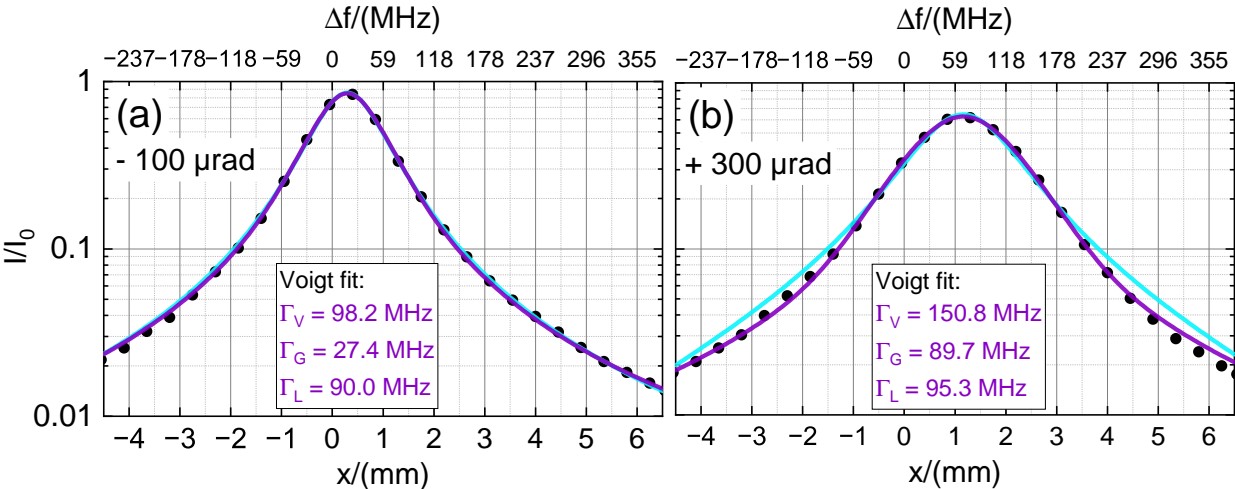

**Figure 8.** Modelled fringes (dots) for FOV = 500 $\mu$rad and x-tilt AOI = $-100$ $\mu$rad (a) and AOI = 300 $\mu$rad (b). Corresponding best-fits of a Lorentzian [Eq. (2)] and a Voigt profile [Eq. (5)] are indicated by the light-blue and purple lines, respectively. The FWHM obtained from the Voigt fit are given by the inset. y-axes are in log-scale to visualize the improved fit of the Voigt profile to the strongly Gaussian broadened fringe in panel b.

The evolution of the respective Lorentzian and Gaussian contributions depending on AOI is illustrated in Fig. 9 for the full set of fringes shown in Fig. 6 b. Notably, the Lorentzian component (magenta) remains almost constant within the range 95 MHz

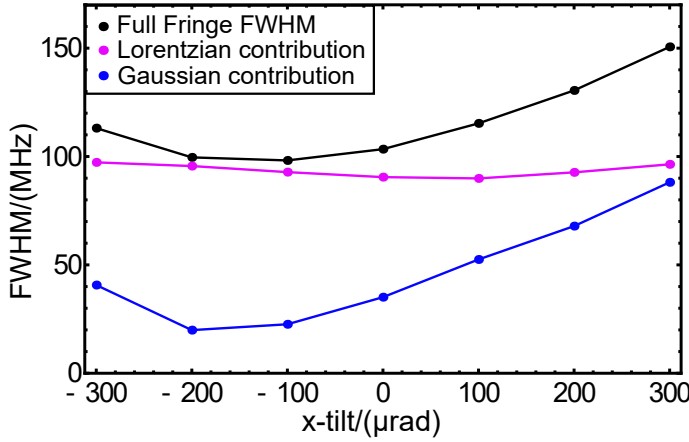

**Figure 9.** The Lorentzian (magenta) and Gaussian (blue) components derived by Voigt profile analysis of the modelled fringes shown in Fig. 6 b.

to 100 MHz FWHM, whereas the Gaussian component (blue) increases from $\approx$ 20 MHz at AOI = $-200$ MHz to 90 MHz for





AOI = 300 $\mu$rad (see also Fig. 8 b). Typically, the error limits on these values are less than $\approx 2$ MHz, but somewhat larger for smaller Gaussian components of $< 30$ MHz.

It is thus clear that the increase in overall fringe width from $\approx 100$ to 150 MHz is due to the increasing Gaussian component at larger AOIs, for the given FOV. Indeed, an AOI approaching 400 $\mu$rad (as evident for the Aeolus Fabry-Perot channel Witschas et al., 2022b) would give a full fringe width of about 175 MHz FWHM, with a Gaussian component somewhat

greater than $\approx 115$ MHz. Note that these values have still not incorporated the laser Gaussian pulse width of 45 MHz, as discussed in the following subsection.

### 4.3 Incorporation of laser pulse into Fizeau profile

The underlying rationale of the present investigation is to develop a more complete physical understanding of the Fizeau fringe and its composition. The two previous subsections have established that, somewhat unexpectedly, the dominant Gaussian nature

of two large contributions – the laser pulse and the impact of the AOI and FOV. There would be every expectation that, on folding these contributions into the complete Fizeau profile, their respective elements would combine together in the usual manner for Gaussians, i.e. by root sum of squares. Nevertheless, it was considered valuable and constructive to investigate this, and to both test the modelling/analytic procedures and promote confidence therein.

The laser pulse was considered as a Gaussian profile with $\Gamma_\mathcal{G} = 45$ MHz, and of unit power. This was convolved in the mod-

elling process with two fringe profiles drawn from Fig. 6 b, with AOI = $-100$ $\mu$rad and AOI = 300 $\mu$rad and FOV = 500 $\mu$rad. The resultant profiles are shown in Fig. 10, with the originals shown in black and the convolved fringe shown in magenta. The y–scale of intensity is normalised against $I_0$ as the incident intensity on the plates, and incorporates a representative value for plate absorption of 0.006 (together with $R = 0.88$). As expected, the slight increase in FWHM, along with the decrease in peak

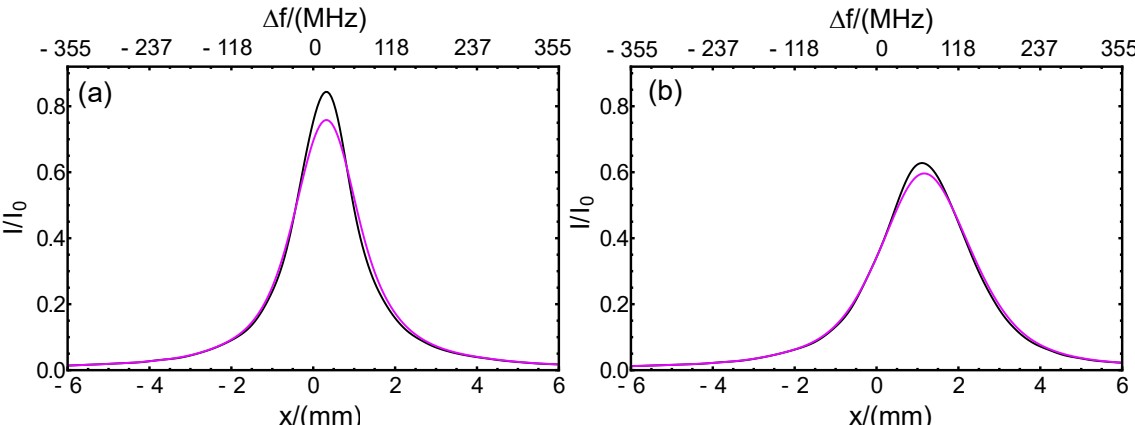

**Figure 10.** Fringe profiles modelled before (black) and after (magenta) convolution with a Gaussian laser pulse profile of 45 MHz FWHM. (a): AOI = $-100$ $\mu$rad, FOV = 500 $\mu$rad. (b): AOI = $+300$ $\mu$rad, FOV = 500 $\mu$rad.

height, for the convolved fringe is apparent.





## 4.4 Plate defects and fringe skewness

Early wave-optic modelling of ideal sinusoidal circular plate defects (±0.5 nm to ±4 nm on both plates) showed cyclic frequency shifts of up to ±2 nm, and gross fringe asymmetry as well as secondary maxima for defects larger than ≈ |2 nm|. Somewhat later, an interferometric mapping of one set of plates became available and showed small-scale cyclic variations (in optical path separation) in the range from ±0.5 nm to ±1.5 nm, which were furthermore overlaid on large scale changes of up to 5 nm.

This measured topography was modelled and a set of 3 representative fringes are shown in Fig. 11. Analysis shows small-

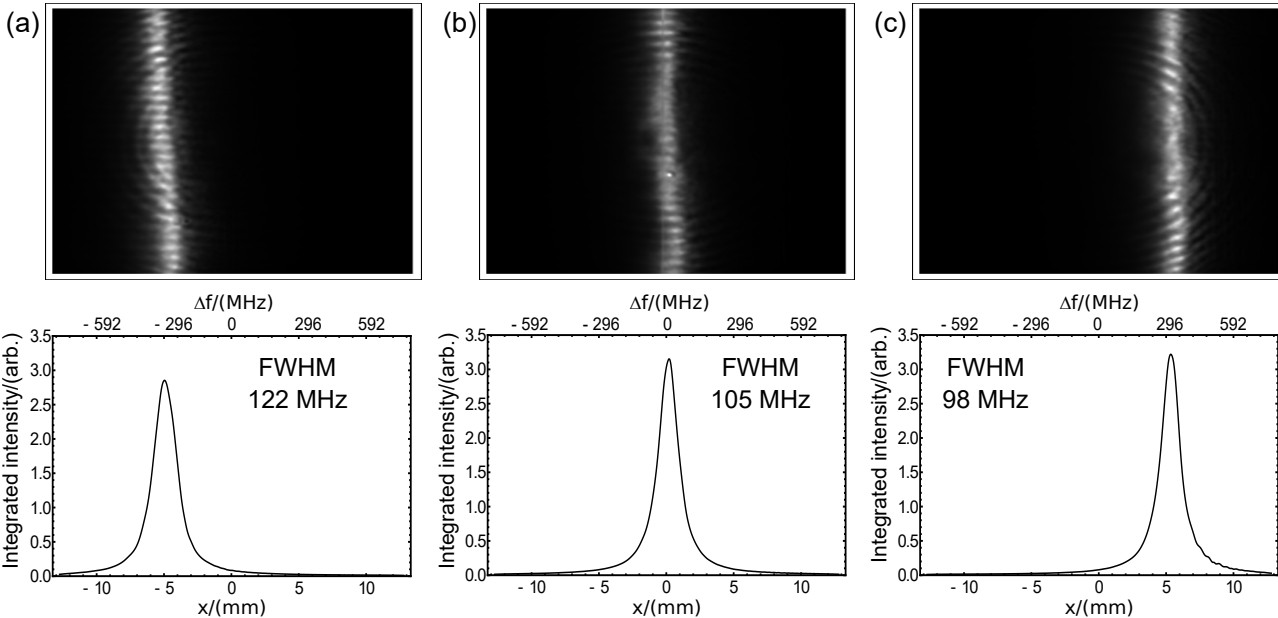

**Figure 11.** Modelled Fizeau fringes using the measured plate topography for a set of plates as shown in the top row images. Note the breakup of the fringes, characteristic of small-scale, semi-regular, groove defects and the fringe tilt (skewness) most evident in fringe (a), attributable to larger scale defects from top to bottom of the plates. The corresponding vertically integrated fringe profiles as well as their FWHM are shown in the lower row.

scale frequency shifts of ±1 MHz, and in addition, larger-scale variations of up to ≈ 14 MHz. On further examination, the two broadened fringes in Fig. 11 are not precisely vertical (i.e. are skewed) with equivalent frequency shifts from top to bottom of ≈ 63 MHz (a) and ≈ 31 MHz (b). It is readily shown that this skewness would give an equivalent width 'top-hat' broadening function, with impact that closely matches the increased widths of profiles (a) and (b) compared with (c). The skewness shift of ≈ 63 MHz is also consistent with a shift of resonant frequency given by $(2\partial d/\lambda) \cdot \Gamma_{\text{FSR}} \approx 60$ MHz, for a large-scale plate separation defect of $\partial d \approx 5$ nm, compared with the plate separation of 68.5 mm. This in turn leads to the fact that the Aeolus Fizeau fringe width changes with wedge position, i.e., with frequency. The impact of the fringe skewness on the wind retrieval of the Aeolus airborne demonstrator was also discussed by Lux et al. (2022).





## 4.5 The impact of Rayleigh-Brillouin scattering

At low levels of aerosol Mie scattering, the signal output from the Fizeau interferometer is increasingly dominated by broadband molecular RB scattering. Such scattering is typically of width in the range from 3.4 GHz to 4.3 GHz FWHM, depending upon altitude, or rather pressure $p$ and temperature $T$, and is of near Gaussian spectral shape (Witschas et al., 2010; Witschas, 2011a, b, 2012). In consequence, the measurement accuracy of Doppler shifts from small aerosol signals is strongly impacted by the broadband background signal for low-level aerosol signals. It is thus important to have good knowledge of the spectral distribution and strength of this background as it appears in the Fizeau output.

Fig. 12 (a) shows a Gaussian profile of representative width 3.8 GHz ($p = 100$ hPa, $T = 274$ K, $\lambda = 355$ nm), to be convolved at mid-order with the Fizeau instrument of 100 MHz FWHM and $\Gamma_{\mathrm{FSR}} = 2.2$ GHz. Full account is taken of the overlap from successive orders at $\pm 1$ FSR, $\pm 2$ FSR, etc. It is instructive to consider this overlap of orders a little further. In Fig. 12 (b), the central peak of zero order (black) is shown with $1^{\mathrm{st}}$ order (purple) and $2^{\mathrm{nd}}$ order (light-blue). Their successive summation is shown in figure Fig. 12 (c), where the purple line indicates the summation of zero and first order, and the light blue line indicates the further addition of the second order. It is important to notice that the resultant background (light blue) is essentially flat with relative intensity of 1.84, compared with the zero-order peak.

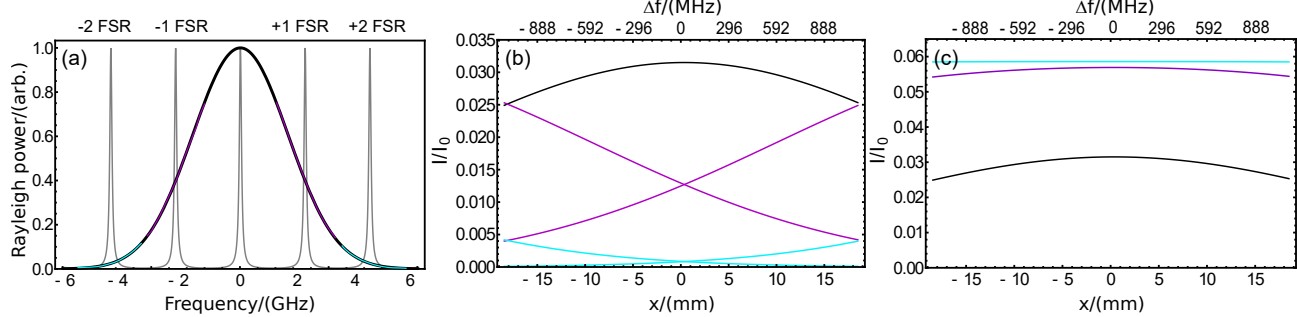

**Figure 12.** (a) Gaussian profile of 3.8 GHz FWHM ($p = 100$ hPa, $T = 274$ K, $\lambda = 355$ nm), as representative of a Rayleigh-Brillouin spectrum, for convolution with the Fizeau instrument function of 100 MHz FWHM and FSR = 2.2 GHz. sketched by the light gray line. (b) Illustration of the overlap across one FSR for successive Gaussian profiles set at the centre, i.e. zero order, (in black), $\pm 1$ FSR (magenta) and $\pm 2$ FSR (light-blue). The respective regions are also highlighted in panel a. (c) Summation of successive orders. For all three orders this is essentially flat (light-blue). Note the intensity ratio for the summed orders is 1.84 times the central order.



## 5 Impacts on the wind measurement accuracy

### 5.1 Impact of line broadening on fringe shift and Doppler wind measurement accuracy

### 5.1.1 Quantum limited accuracy and SNR analysis of fringes

We consider, first, the accuracy of frequency estimation in the case of a fringe where there is no background light and the only noise source is shot noise on the signal photoelectrons. The following result for the standard deviation of frequency estimates $\delta f$ was derived by (Vaughan, 1989, Appendix 10)

$$\delta f = \frac{C \cdot \Delta f}{\langle N_S \rangle^{1/2}}, \tag{9}$$

where $\Delta f$ is the FWHM of the ultimate signal profile emerging from the instrument before detection, and the term in the denominator is the square root of the signal energy within the profile, expressed as the mean number of electron-counts $\langle N_S \rangle$ for a measurement ensemble. $C$ is a constant of order 1, which depends on the actual spectral shape of the fringe. The derivation of Eq. (9) was based on determining the median position of the fringe (i.e. with equal numbers of photodetections on either side).

By doing so, the constant $C$ was shown to be $C_{\mathcal{L}} = \pi/4 \approx 0.785$ for a Lorentzian profile, and $C_{\mathcal{G}} = [\pi/(16 \cdot \ln 2)]^{1/2} \approx 0.532$ for a Gaussian profile [see also equations A94 and A95 in (Vaughan, 1989)]. Note that there is an error in this reference whereby the values given are greater than they should be by a factor of $\sqrt{2}$, as only one part of the fringe was considered for the derivation. The correct values are the ones we use here. The constant for a Voigt-shaped fringe $C_{\mathcal{V}}$ lies between the two values given above, depending on the respective Lorentzian and Gaussian contributions.

Now, Eq. (9) has the same functional form as the Cramér-Rao lower bound (CRLB) for this frequency estimation scenario, the only difference being the value of the multiplying constant. The CRLB is a value of the standard deviation which cannot be bettered by any unbiased frequency estimation method. The CRLB for a Gaussian profile is given in Rye (1998) by their Eq. (18). After converting their Gaussian $1/e^2$-radius to a FWHM, it is found that the CRLB multiplying factor is $0.425$. If one does the same calculation for a Lorentzian profile the result is a multiplying factor of $\sqrt{2} \approx 0.707$. It is not surprising that

these multiplying factors are somewhat lower than those given by Vaughan (1989), since the former represent an ultimate limit and the latter come from an analysis of an actual frequency estimation algorithm. We use the latter approach here and, as will be seen, find good agreement with frequency estimation using least squares fitting to a defined profile.

In the ideal formulation given by Eq. (9), it is supposed that the electron-counts $N_S$ are free of spurious noise, dark current and additional background signal. Consequently, the Poisson quantum-limited noise for the mean signal is equal to $\langle N_S \rangle^{1/2}$.

In Eq. (9), the final term may be usefully considered as the SNR of the system, i.e.:

$$\text{SNR} = \frac{\langle N_S \rangle}{\langle N_S \rangle^{1/2}} = \langle N_S \rangle^{1/2}, \tag{10}$$

and hence, Eq. (9) can be transformed to

$$\delta f = \frac{C \cdot \Delta f}{\text{SNR}}. \tag{11}$$





Inspection of Eqs. (9) and (11) implies the paramount importance of the spectral line width $\Delta f$ for the measurement accuracy
and the evaluation of $\delta f$. As a simple example, a 2-fold spectroscopic reduction in $\Delta f$ would be equivalent, in terms of accuracy, to a 4-fold energy increase in $\langle N_S \rangle$, requiring either an increase of the telescope diameter by a factor of 2, or an increase of the laser pulse energy by a factor of 4.

### 5.1.2  Fizeau fringe modelling and simulation

In Fig. 13 (a), two Fizeau fringes following an ideal Lorentzian profile according to Eq. (2) with $\Gamma_{\mathcal{L}} = 100\,\mathrm{MHz}$ (1.69 mm) are
shown. A total of 800 photo electrons (black dots), and 3200 photo electrons (light blue dots) are distributed across 512 sampling points. The number of 512 points was chosen to give a much larger number of pixels across the fringe profile, so that the results can be compared with expressions such as Eq. (9), which are derived on the assumption of negligibly-small pixels. The mean number of photo electrons is proportional to the fringe profile shown by the solid line. The sample contents, i.e. the ordinates $N_S$, are random numbers taken from generated Poisson distributions. The centre frequency and height of the profile were used as fitting parameters. The statistical variations in the centre frequency estimate were investigated for 1000

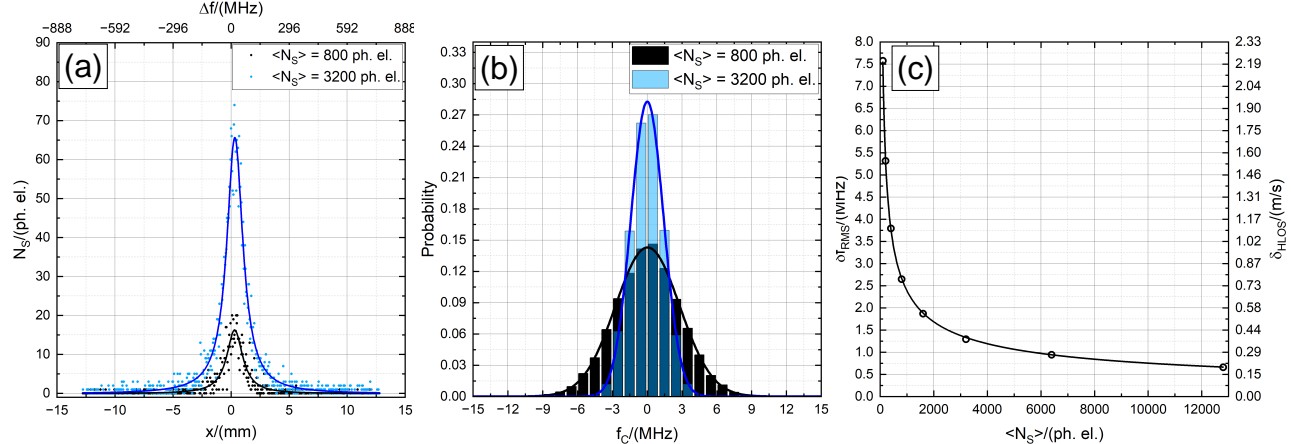

**Figure 13.** (a) Example of two Fizeau fringes (ideal Lorentzian profile) with $\Gamma_{\mathcal{L}} = 100\,\mathrm{MHz}$ and shot noise (points). A total of 800 photo electrons (black dots), and 3200 photo electrons (light blue dots) are distributed across 512 sampling points, and the solid lines indicate corresponding best-fits of a Lorentzian. (b) Histogram for 1000 different realisations of centre frequency for the individual fitted Lorentzians. (c) Root-mean-square of the frequency estimates given for the frequency (left y-axis) and HLOS wind speed (right y-axis). The black line indicates a best-fit of Eq. (9) to the data ($\Delta f = 100\,\mathrm{MHz}$).


different realisations of the Poisson–distributed shot noise. The resultant histograms, with bin widths of 1.5 MHz, are shown in Fig.13 (b). The zero-frequency point is taken to be the centre of the fringe. Note that the plotted histogram is a discretised version of the probability density, where the probability of a value lying within a given histogram bin is the height of the bin multiplied by its width. Thus, the width of the histogram gives a measure of the accuracy of the frequency estimates. For the
shown data, the root-mean-square (RMS) and hence, the standard deviations of these frequency estimates is 2.7 MHz (black,



800 photo electrons) and 1.4 MHz (light-blue, 3200 photo electrons), which corresponds to a wind velocity error in horizontal LOS (HLOS) direction of $0.79\,\mathrm{m\,s^{-1}}$ and $0.41\,\mathrm{m\,s^{-1}}$, considering the conversion of $1\,\mathrm{m\,s^{-1}}$ HLOS wind speed to $3.43$ MHz frequency shift, as resulting from the Doppler equation and considering a off-nadir angle of $37.6°$.

This calculation, with 1000 estimations per set, was repeated for eight different values of the mean number of electron counts $\langle N_S \rangle = (1,2,4,8,16,32,64,128)\cdot 10^2$. The RMS values $\delta f_{\mathrm{RMS}}$ of the eight resultant histograms of fringe centre (per Fig.13 c) were closely proportional to $\langle N_S \rangle^{-1/2}$. Expressed in terms of fringe width, $\Delta f$, the best fitted curve according to Eq. (9) (Fig.13 c, black line) yields a $C_\mathcal{V}$ value of $0.788$, which is in close agreement with the numerical value for a Lorentzian profile ($0.785$) as mentioned above.

## 5.2 Impact of Rayleigh-Brillouin background signal on the SNR and the measurement accuracy

### 5.2.1 Fringe simulation and modelling with significant background

In reality, the aerosol Mie peak in the Fizeau will sit on top of a pedestal of Rayleigh background. Even if this background is entirely uniform, shot noise on the background photo electrons will degrade the performance of the fringe measurement. This situation was modelled by adding a flat pedestal to the fringe pattern and then calculating mean photo-electron numbers and shot noise realisations as before. The size of the pedestal was characterised by the mean number of background photo electrons

$N_{\mathrm{ped}}$ per pixel column. With 16 columns across the detector, the total number in the background is obviously $16 \times N_{\mathrm{ped}}$. The simulation analysis includes the impact of the detector pixelation and the frequency estimation algorithm was modified to account for a background pedestal of unknown height. The number of realisations used to analyse the statistics was increased to 10000. This early investigation was made as realistic as possible by selecting fringe parameters close to those expected for Aeolus at the time. These values have now largely been superseded, but the study itself proved very instructive and produced

valuable guidelines for much of the following work.

The modelled Fizeau fringe was of width $\Delta f = 158.7$ MHz, made up of a Lorentzian with $\Gamma_\mathcal{L} \approx 148$ MHz (close to the expected sum of Fizeau instrumental and laser pulse width), and a small Gaussian component, due to FOV speckle broadening, estimated at $\Gamma_\mathcal{G} \approx 25$ MHz.

The main signal $N_{\mathrm{S}}$ was set at 1600 photo electrons, and a wide range of $N_{\mathrm{ped}}$ up to 6750 photo electrons per detector column

were examined. Fig. 14 a shows a set of 10000 frequency estimates for the pedestal $N_{\mathrm{ped}} = 6400$ photo electrons (black) and for $N_{\mathrm{ped}} = 1600$ photo electrons (blue). In panel b, the corresponding histograms are shown in sets of width $5\,\mathrm{m\,s^{-1}}$ (bars); the blue and black curves indicate respectively the Gaussian fits, establishing that the frequency estimates are close to a normal distribution. In panel c, the standard deviations of the frequency estimates for 10 different pedestal levels are depicted. Note that for $N_{\mathrm{ped}} = 1600$, the simulated st.dev. value of small $\delta f_{\mathrm{sim}} = 7.7$ MHz is equivalent to an HLOS Doppler velocity accuracy

of $\delta_{v_{\mathrm{HLOS}}} \approx 2.2\,\mathrm{m\,s^{-1}}$. For $N_{\mathrm{ped}} = 6400$, the simulated st.dev. value of small $\delta f_{\mathrm{sim}} = 13.8$ MHz is equivalent to an HLOS Doppler velocity accuracy of $\delta_{v_{\mathrm{HLOS}}} \approx 4.0\,\mathrm{m\,s^{-1}}$. This means that background signals of this order could in principle explain the random error as obtained for Aeolus. However, the actual $N_{\mathrm{ped}}$ levels are in fact more than 40 times smaller than the large 6400 photo electrons considered in this example.

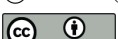



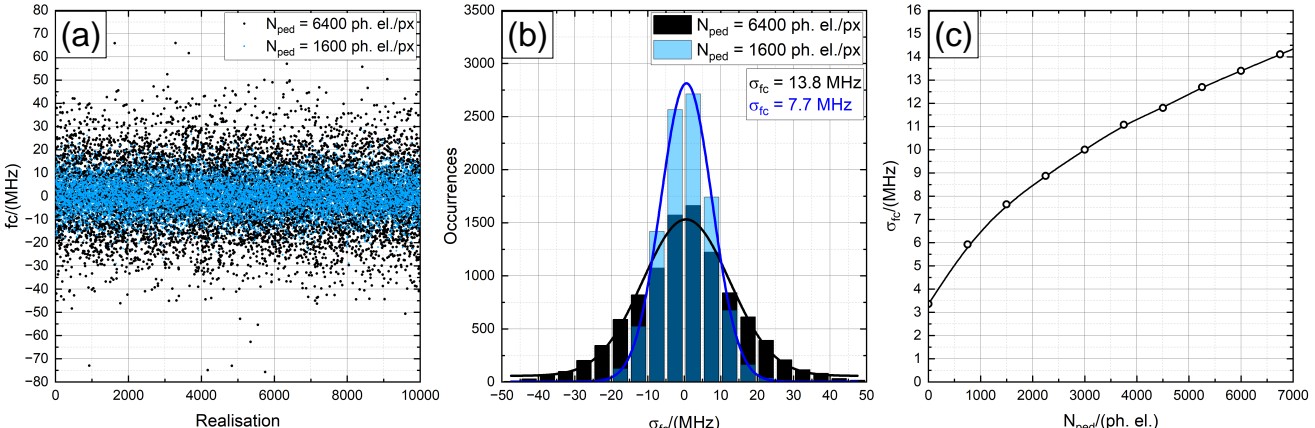

**Figure 14.** (a) Set of 10000 frequency estimates for a pedestal of 6400 background photo electrons per detector column (black) and of 1600 background photo electrons per detector column (blue) simulated with a fringe of width $\Delta f = 158.7\,\mathrm{MHz}$ and a signal level of $N_\mathrm{S} = 1600$. (b) Corresponding histogram of the data shown in panel a (bars) and related Gaussian fits (lines). The standard deviation of the data is indicated by the inset. (c) Standard deviation of frequency estimates versus mean number of pedestal photo electrons per detector column. The mean number of signal photo electrons for all cases is 1600.

### 5.2.2 Fringe calculations

This section presents an analytical framework that seeks to complement the extensive simulation and computations of the previous sections, as illustrated in Fig. 13 and Fig. 14. When there is a significant background pedestal present, a very basic SNR may be simply defined as:

$$\mathrm{SNR}_\mathrm{basic} = \frac{N_\mathrm{S}}{(N_\mathrm{S} + m \cdot N_\mathrm{ped})^{1/2}}, \tag{12}$$

with $m$ set equal to the total number of detector channels ($m = 16$), the noise term (denominator) in the bracket is the total 570 number of photoelectrons recorded across the detector. Simple calculation of Eq. (12), comparable to the simulations and computation of Fig. 14, with similar large values of $N_\mathrm{ped}$, illustrates its relative crudity. In particular, for $\langle N_S \rangle = 1600$ ph.el. and $N_\mathrm{ped} = 1600$ ph.el., $\mathrm{SNR}_\mathrm{basic} = 9.7$, equivalent to a error of $\delta f = 12.4\,\mathrm{MHz}$. For the case with the even larger background signal of $N_\mathrm{ped} = 6400$ ph.el., $\mathrm{SNR}_\mathrm{basic} = 5.0$, equivalent to an error of $\delta f = 24.2\,\mathrm{MHz}$.

These values are notably larger errors than those values shown in Fig. 14 b of 7.7 MHz and 13.8 MHz, respectively. On 575 reflection, this is hardly surprising: the bulk of the signal is contained within a small number of central channels, while the outer channels are dominated by the pedestal noise background. This simply illustrates the well-known spectroscopic principle of minimising any analytic or search bandwidth $f_\mathrm{AB}$ in order to maximise SNR and improve signal accuracy. The question, then is, what analytic formulation of SNR would provide accuracy values closer to those of the simulation and computation





procedures of the previous sections? A physically reasonable refined SNR, for insertion in Eq. (11) is:

$$\mathrm{SNR}_{\mathrm{refined}} = \frac{k_{\mathrm{r}} \cdot N_{\mathrm{S}}}{\left(k_{\mathrm{r}} \cdot N_{\mathrm{S}} + n \cdot N_{\mathrm{ped}}\right)^{1/2}}, \qquad (13)$$

where $k_{\mathrm{r}}$ is the fraction of $N_{\mathrm{S}}$ contained within an effective analytic bandwidth $f_{\mathrm{AB}}$, selected for purposes of calculation as $f_{\mathrm{AB}} = r \cdot \Delta f = n \cdot \Gamma_{\mathrm{TH}}$, with $n$ being the number of pixels covered by the analytical bandwidth. Hence, $r$ is the ratio of the analytical bandwidth $f_{\mathrm{AB}}$ and the fringe width $\Delta f$. It is worth mentioning that the SNR calculation by means of Eq. (13) differs from the one used in the Aeolus processor. However, it provides a good method for investigating the Mie wind performance

evolution for varying instrumental parameters as it is shown in the following.

Calculated values of $\mathrm{SNR}_{\mathrm{refined}}$ obviously depend on the selected values of $r$. For best accuracy, the optimum choice would provide the largest possible $\mathrm{SNR}_{\mathrm{refined}}$, using the experimentally defined values $N_{\mathrm{S}}$, $N_{\mathrm{ped}}$ and $\Delta f$. By inserting typical Aeolus parameters, it is readily shown for $N_{\mathrm{S}} \approx N_{\mathrm{ped}}$, that plots of $\mathrm{SNR}_{\mathrm{refined}}$ versus $r$ exhibit a broad profile, typically peaking in the range $r \approx 1.5$ to $r \approx 2.3$, with $\mathrm{SNR}_{\mathrm{refined}}$ remaining within $\approx 4\%$ of the maximum over the much broader range ($r = 1.2$

to $r = 3.0$).

The utility of $\mathrm{SNR}_{\mathrm{refined}}$ for relatively large $N_{\mathrm{ped}}$ is simply demonstrated in reference to the data of Fig. 14. As noted, this profile of 158.7 MHz (FWHM) has a small Gaussian component. It is estimated that approximately $90\%$ of the full Voigt width is due to the much larger Lorentzian component. Extensive analysis and modelling shows that the equivalent $C$ coefficient results in $C_{\mathcal{V}} = 0.755$, for this case; a near optimum value of $r$ is $\approx 1.6$ i.e., an analytic bandwidth of $\approx 250$ MHz, equivalent

to $n = 2.5$ for $\Gamma_{\mathrm{TH}} = 100$ MHz. The resultant value of $k_{\mathrm{r}}$ is $\approx 0.67$.

Insertion of these values in Eqs. (11) and (13) gives $\mathrm{SNR}_{\mathrm{refined}} = 15.0$ equivalent to $\delta f = 8.0$ MHz ($N_S = 1600$ ph.el. and $N_{\mathrm{ped}} = 1600$ ph.el.) and $\mathrm{SNR}_{\mathrm{refined}} = 8.2$ equivalent to $\delta f = 14.6$ MHz ($N_S = 1600$ ph.el. and $N_{\mathrm{ped}} = 6400$ ph.el.). Obviously these analytic values of $\delta f$ in Eq. (13) are much closer to the simulated values shown in Fig. 14 than those due to $\mathrm{SNR}_{\mathrm{basic}}$ shown in Eq. (12). The small residual discrepancies of $\approx 5\%$ is readily accounted for by uncertainty in the precise

Lorentzian fraction. For a better comparison, the resulting $\mathrm{SNR}_{\mathrm{basic}}$ and $\mathrm{SNR}_{\mathrm{refined}}$ as well as the corresponding error for the two cases discussed in Fig. 14 are summarized in Table 2.

**Table 2.** Comparison of derived $\mathrm{SNR}_{\mathrm{basic}}$ [Eq. (12)] and $\mathrm{SNR}_{\mathrm{refined}}$ [Eq. (13)] and corresponding $\delta f$ [Eq. (12)] for different $\langle N_S \rangle$ and $N_{\mathrm{ped}}$ values ($\Delta f = 158.7$ MHz, $C_{\mathcal{V}} = 0.755$, $m = 16$, $n = 2.5$, $k_{\mathrm{r}} = 0.67$).

| | $\mathrm{SNR}_{\mathrm{basic}}$ | $\delta f$ | $\mathrm{SNR}_{\mathrm{refined}}$ | $\delta f$ | $\delta f$ from Fig. 14 |
|---|---|---|---|---|---|
| $\langle N_S \rangle = 1600$ ph.el., $N_{\mathrm{ped}=1600 \text{ ph.el.}}$ | 9.7 | 12.4 MHz | 15.1 | 8.0 MHz | 7.7 MHz |
| $\langle N_S \rangle = 1600$ ph.el., $N_{\mathrm{ped}=6400 \text{ ph.el.}}$ | 5.0 | 24.2 MHz | 8.2 | 14.6 MHz | 13.8 MHz |

The foregoing analytic procedure, which will be described in detail and applied in a forthcoming publication, offers a relatively simple, easily calculated representation of the quantum limited accuracy for direct detection spectroscopic systems. It thus provides a useful metric for comparison of potential performance for variation of instrumental parameters. However,

caution is needed in comparing different analytic techniques, via their apparent SNRs.





### 5.3 Measurement accuracy with the Aeolus Fizeau interferometer and potential improvements for future applications

#### 5.3.1 Aeolus experimental performance

For over 4 years in orbit, the Fizeau instrument, primarily intended as a technical demonstrator, has in fact provided an enormous volume of wind data of great value for meteorological analysis (Rennie et al., 2021; Rennie and Isaksen, 2024). The principle experimental main findings of the Aeolus Fizeau system throughout its operation may be summarised as follows.

1. The estimated precision of the HLOS winds varied considerably during the mission and with geolocation, season, processing software version and range-bin settings, particularly for the Rayleigh-clear HLOS winds, but to a much smaller extent for the Mie-cloudy winds with random errors varying from $2.5\,\mathrm{m\,s^{-1}}$ to $3.6\,\mathrm{m\,s^{-1}}$ (HLOS) on horizontal scales of about $10\,\mathrm{km}$ to $20\,\mathrm{km}$ (Rennie and Isaksen, 2024). It is worth noting, that HLOS winds are the LOS winds projected to horizontal direction. Considering the Aeolus pointing angle of $37.6°$, $1\,\mathrm{m\,s^{-1}}$ HLOS wind corresponds to $3.43\,\mathrm{MHz}$ Doppler frequency shift.

2. A significant fraction of the valid Mie-cloudy wind data resulted from strong backscatter from ice and water clouds, including cloud top and more diffuse thin clouds.

3. There were few measurements that may be attributed unequivocally to purely aerosol backscattering, and these were almost entirely due to rare high-backscatter events caused, for example by incipient cirrus cloud formation, volcanic eruption, dust plumes, and wild fire smoke swept high into the atmosphere.

4. There were almost no observations of Mie winds with errors below $1\,\mathrm{m\,s^{-1}}$ (HLOS), in contrast to the expectations from pre-launch simulations and specifications (ESA, 2016).

5. There were only small changes in performance for Mie-cloudy winds when switching between laser flight model A (FM-A) operation and laser FM-B, in contrast to the larger changes that were obvious for Rayleigh-clear winds due to the changing atmospheric path signal levels when operating with each of the lasers.

No clear reasons have been advanced for these discrepancies but they appear to point to a considerable loss of radiometric performance. Conceivably, this might be due to loss of optical alignment accuracy, and reduction of light signal through the optical train, for instance if the field-stop aperture is not positioned at the centre of the optical focus of the telescope, leading to an over-illumination of the field stop. Indeed, this hypothesis would be supported by the large apparent AOI, of order $300\,\mu\mathrm{rad}$ to $400\,\mu\mathrm{rad}$ needed to explain the large spectral line widths discussed in Sect. 3. However, also a larger beam diameter in the field stop due to larger wave-front errors of optics (e.g. the telescope) leads to over-illumination.

#### 5.3.2 Calculation and analysis of the present Aeolus Fizeau performance

**a) Operational Fizeau parameters for fringe analysis**

From the earlier discussion, the operational Aeolus Fizeau fringe may be considered as having an FWHM of $\Delta f = 175\,\mathrm{MHz}$





(before detection), made up of the Fizeau instrumental function (at large AOI and FOV) with components Lorentzian $\approx 95$ MHz and Gaussian $\approx 107$ MHz, folded with the laser pulse profile of Gaussian $\approx 45$ MHz. This gives an $L_{\text{fraction}} = 95/175 = 0.54$, which leads to $C_{\mathcal{V}} \approx 0.66$. An analytic bandwidth $f_{\text{AB}} = 300$ MHz (i.e. $n \approx 3$ pixels) is close to optimum (within $\approx 5\%$), for

significant $N_{\text{ped}}$, and gives the ratio $r = f_{\text{AB}}/\Delta f \approx 1.7$. This leads to a collection efficiency for the Mie signal of $k_r \approx 0.80$. For calculation and comparison in Eqs. (11) and (13), these values are considered as reasonably representative of Aeolus operation and are used in the following analysis.

**b) Analysis of strong signal Aeolus fringes**

A group of strong signal fringes and associated data tables are shown in Fig. 15. These three sets of atmospheric observations

were recorded on 1 June 2022 at around 06:00 UTC, each consisting of five measured Fizeau fringes. The fringes were accumulated over successive paths with horizontal integration length of about 12 kilometres, vertical integration length of 0.75 km, and at different altitudes: (a) 8.45 km (range gate 12), (b) 4.7 km (range gate 17), and (c): 1.7 km (range gate 21). It must be appreciated that these sets have been taken as an example from the many millions of observations recorded in the 4 years of operation, and as such, they are better considered as "indicative" rather than "representative". Examination of

these fringes and associated data reveals two important points. First, the mean value of $N_{\text{S}}$ is of the order of 1000 LSB, which corresponds to $\approx 1462$ ph.el., with the radiometric gain of 0.684 LSB/ph.el. for the Mie ACCD detector (Lux et al., 2024). Hence, the fringes are based on a strong signal but fluctuate significantly, ranging from 0 LSB to 2500 LSB across the 15 spectra. Second, the mean value of $N_{\text{ped}}$ is $\approx 40$ LSB ($\approx 58$ ph.el.), which is more than an order of magnitude lower than some of the values calculated and modelled in the previous section; it is also significantly lower than anticipated for the

Rayleigh-Brillouin background. This evidence offers further support for the hypothesis of reduced radiometric efficiency in Aeolus operation. However, for single realizations, it can also happen that the cloud top appears at the top of the range bin, which would result in a low Rayleigh background as well.

From the above data 12 of the 15 fringes have $\text{SNR}_{\text{refined}} > 25$ and 3 have $\text{SNR}_{\text{refined}} \approx 50$, corresponding to standard deviation accuracies (per signal statistics) of $\delta_{v_{\text{HLOS}}} \approx 1.4 \text{ m s}^{-1}$ and $\delta_{v_{\text{HLOS}}} \approx 0.7 \text{ m s}^{-1}$, respectively. It is worth noting that

for set a), at high altitude 8.45 km, the spread in fringe centre $x_c$ is only 0.041 px (4.1 MHz), equivalent to a range of measured HLOS velocities of $\approx 1.2 \text{ m s}^{-1}$, close to the statistical value. In contrast, for set b), the spread of 0.207 px, equivalent to a range of measured HLOS velocities of $\approx 6.0 \text{ m s}^{-1}$; this might suggest increased shear and turbulence along the 70 km atmospheric path, at this lower altitude of 4.7 km. It is further worth noting that the variation on the values of the derived $N_{\text{ped}}$ of 36.4 LSB to 41.6 LSB, in set a), are within $\pm 2$ times the standard deviation (per Poisson statistics) of the mean value of

37.9 LSB. In contrast, the variation of $N_{\text{ped}}$ at the lower altitudes b) and c) are well outside the Poisson values; this is probably due to variable levels of attenuation in the layers above.

In summary, these 12 measurements with notably strong signals suggest that in these cases the scattering is dominated by clouds. The fact that it is evident at all altitudes could indicate that the clouds are sufficiently thin and diffuse to permit adequate transmission to lower altitudes. As the three examples stem from three different observations/profiles of the orbit. There could

have been a thick cloud in different altitudes for each of the examples. However, it is more likely that it is largely due to scattered clouds of low overall coverage over the 12 km path per measurement.

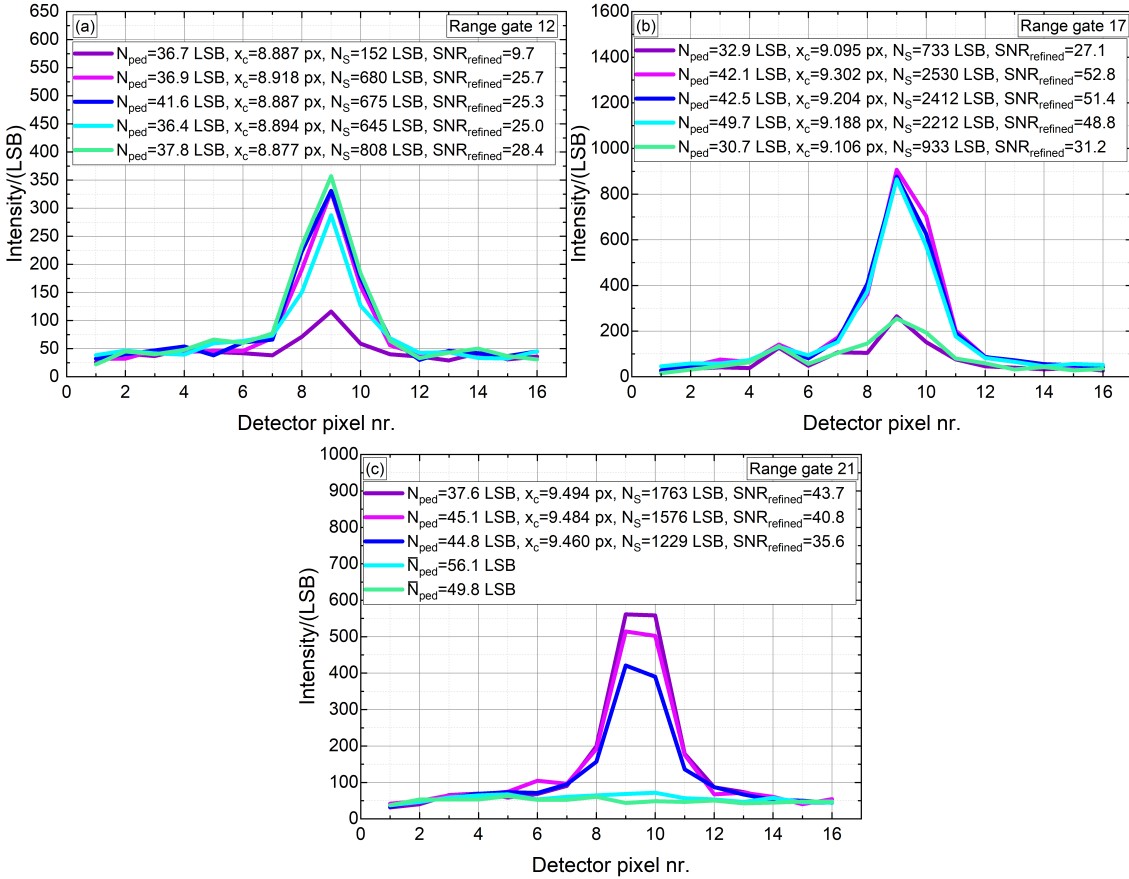

**Figure 15.** Three sets of five atmospheric fringe profiles recorded by Aeolus on 1 June 2022 at 06:00 UTC at different altitudes of 8.8 km corresponding to range gate 12 (a), 4.7 km (range gate 17, b) and 1.7 km (range gate 21, c). Derived data tables containing values of $N_{\text{ped}}$ (per pixel), the center position in pixels $x_c$, $N_{\text{S}}$, as well as the $\text{SNR}_{\text{refined}}$ [Eq. (13)] using $\Delta f = 175\,\text{MHz}$, $C_{\mathcal{V}} = 0.66$, $k_r \approx 0.80$, and $n = 3\,\text{px}$ are given for each fringe by the respective inset.

**c) Analysis of weak signal Aeolus fringes**

Examination of Aeolus data reveals that $\approx 50\%$ of the valid Mie winds from the Aeolus processor are retrieved with notably smaller SNR between $8.5$ and $15$. Note that this SNR based on the Mie core 2 algorithm, is derived somewhat differently and

employs a Lorentzian fit. Hence, the values differ slightly from the $\text{SNR}_{\text{refined}}$ values presently employed.

    For present purposes, 5 indicative low signal fringes have been taken from the same orbit as used in the previous subsection and are shown in Fig. 16. The associated data table shows smaller values of $\text{SNR}_{\text{refined}}$ ranging from 16.0 down to 12.4. These are equivalent to standard deviation values of $\delta f = 7.2\,\text{MHz}$ and $\delta f = 9.3\,\text{MHz}$, respectively, equivalent to $\delta_{v_{\text{HLOS}}} = 2.1\,\text{m\,s}^{-1}$ and $\delta_{v_{\text{HLOS}}} = 2.7\,\text{m\,s}^{-1}$. These values are indeed somewhat smaller than the range of errors of $2.5\,\text{m\,s}^{-1}$ to $3.6\,\text{m\,s}^{-1}$ noted

for Mie cloudy winds (Rennie and Isaksen, 2024).



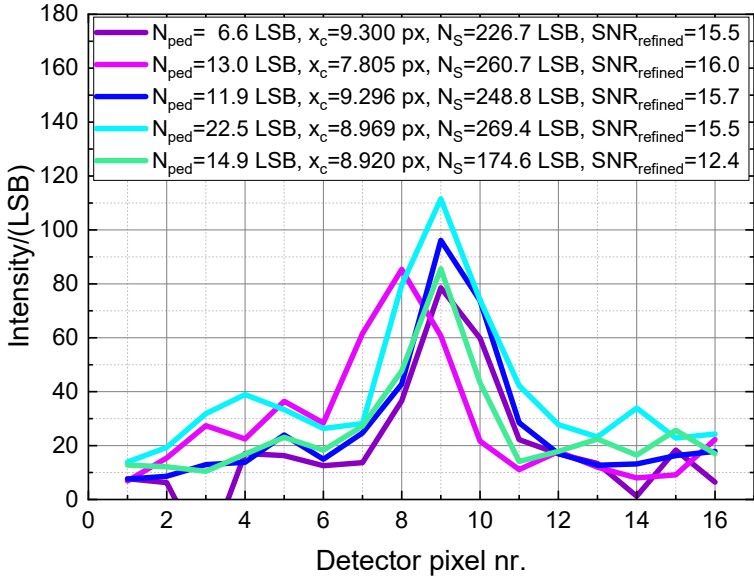

**Figure 16.** Five atmospheric fringe profiles recorded by Aeolus on 1 June 2022 at 06:00 UTC with an SNR varying between 10.4 and 15. The figure label contains values of $N_{\text{ped}}$, the center position in pixels $x_c$, $N_S$ determined by a best-fit of Eq. (5), the $\text{SNR}_{\text{refined}}$ [Eq. (13)] using $\Delta f = 175$ MHz, $C_\mathcal{V} = 0.66$, $k_r \approx 0.80$, and $n = 3$ px are given for each fringe.

Examination of the spectra in Fig. 16 indicates that, in spectroscopic terms, these are still quite well defined fringes. It is generally considered that, for a semi-ideal stable system, a well defined SNR greater than $\approx 6$ provides a reliable, statistically "valid" measurement, although the equivalent large error may make it "not useful for purpose". In the present case an $\text{SNR}_{\text{refined}}$ of 9 would lead to $\delta_{v_{\text{HLOS}}} = 3.6 \ \text{m s}^{-1}$, at the limit of acceptability and usefulness.

## 5.4 Potential improvements to the Fizeau measurement accuracy in short and long term

During this investigation, it became apparent that technical and parametric changes to the Fizeau instrument would notably improve the wind measurement accuracy. The principal changes may be briefly summarised as the reduction of the actual fringe profile width as it emerges from the Fizeau, a correction of the radiometric efficiency (RME) signal loss factor as it was existing for Aeolus, as well as the introduction of additional optical pre-filtering to further reduce both Rayleigh-Brillouin and solar background. The following notes provide a basic outline of a potential accuracy improvement.

For simple comparison, the example of a rather weak signal, with accuracy $\delta_{v_{\text{HLOS}}} = 3.8 \ \text{m s}^{-1}$ at the outer limit of useful range, has been selected, and hence slightly worse than what would be expected from the fringes shown in Fig. 16. As it is shown in the following, potential improvements in this value are then demonstrated to achieve notably better than $\delta_{v_{\text{HLOS}}} = 2.0 \ \text{m s}^{-1}$, and in future upgraded systems, better than $\delta_{v_{\text{HLOS}}} = 1.0 \ \text{m s}^{-1}$.





### 5.4.1 Potential improvements within the framework of the Aeolus Fizeau parameters

In the following, four different scenarios a) to d) with different Fizeau parameters are discussed regarding their corresponding wind accuracy.

#### a) Per the operational Aeolus instrument

Consider the operational Aeolus Fizeau line width $\Delta f = 175$ MHz (at AOI $\approx 300$ $\mu$rad) with $N_\mathrm{S} = 140$ LSB and background $N_\mathrm{ped} = 30$ LSB/px, equivalent to about $N_\mathrm{S} = 204.7$ ph.el. and $N_\mathrm{ped} = 43.9$ ph.el./px, considering the ACCD radiometric gain of $0.684$. For the present and all following calculations of SNR and $\delta f$ [per Eqs. (9) and (13)], a mid-range set of representative parameters of $C_\mathcal{V} = 0.7$, $k_\mathrm{r} = 0.8$ and $r = 1.8$ (defining the analytic bandwidth to be $r \cdot \Delta f$) have been selected. Insertion of these values leads to $\mathrm{SNR}_\mathrm{refined} = 9.4$, $\delta f = 13.0$ MHz, and $\delta_{v_\mathrm{HLOS}} = 3.8 \, \mathrm{m \, s^{-1}}$.

Taking account of the RME loss factor (for both $N_\mathrm{S}$ and $N_\mathrm{ped}$), current expectation suggests this could lie in the range 2 to 3. For the present calculation, suppose a loss factor of 2.5. It is simply shown that the increase in $\mathrm{SNR}_\mathrm{refined}$ is $2.5^{(1/2)} = 1.58$. Hence, in the above values, the resultant accuracies become $\delta f = 8.2$ MHz and $\delta_{v_\mathrm{HLOS}} = 2.4 \, \mathrm{m \, s^{-1}}$.

As a further step, consider optical pre-filtering applied to the input beam of the Fizeau interferometer to further reduce background and eliminate overlap of successive orders of R-B scattering. If completely successful, this would have minimal effect on $N_\mathrm{S}$, while reducing $N_\mathrm{ped}$ by a factor of $\approx 1.84$. Further calculation leads to moderately improved $\delta f = 7.3$ MHz and $\delta_{v_\mathrm{HLOS}} = 2.1 \, \mathrm{m \, s^{-1}}$. These 3 sets of values are indicated in column (a) of Fig. 17, where the initial situation is indicated by the purple line, and the improvements for correction of the RME and optical pre-filtering are indicated by the dark-blue and light-blue line, respectively.

#### b) Operate at the optimum AOI

With controlled operation at an optimum AOI of $\approx -100$ $\mu$rad to realize the minimum spectral line width (i.e. the so-called 'sweet spot'), the $\Delta f$ is reduced to $\approx 115$ MHz. As shown in Fig. 6, the signal energies $N_\mathrm{S}$ and $N_\mathrm{ped}$ for different AOIs are essentially unchanged. Insertion of these values in Eqs. (9) and (13) leads to $\mathrm{SNR}_\mathrm{refined} = 10.3$, $\delta f = 7.8$ MHz, and $\delta_{v_\mathrm{HLOS}} = 2.3 \, \mathrm{m \, s^{-1}}$. Note the small increase in SNR due to the reduced background signal (with smaller analytic bandwidth) entering the equations. Compensation for the RME loss factor gives the accuracy values $\delta f = 5.0$ MHz and $\delta_{v_\mathrm{HLOS}} = 1.5 \, \mathrm{m \, s^{-1}}$. With further incorporation of optical pre-filtering, the values become $\delta f = 4.5$ MHz and $\delta_{v_\mathrm{HLOS}} = 1.3 \, \mathrm{m \, s^{-1}}$. These 3 sets of values are indicated in column (b) of Fig. 17.

### 5.4.2 Potential performance of upgraded Fizeau systems with optimised parameters

Any upgraded system (for Aeolus-type operation) must incorporate two vital considerations. Firstly, the meteorological specification requires a wind velocity measurement capability of up to $\pm 100 \, \mathrm{m \, s^{-1}}$ in HLOS direction ($v_\mathrm{HLOS}$). This is typically equivalent to a $v_\mathrm{LOS}$ extending to $\pm 61 \, \mathrm{m \, s^{-1}}$, considering a off-nadir angle of $37.6°$. Secondly, it is of paramount spectroscopic importance to maximise and maintain the signal collection of narrow-band at atmospheric Mie scattering. This necessarily requires careful consideration of spectroscopic factors of frequency dispersion (MHz per mm) at the plates (and equivalent fringe

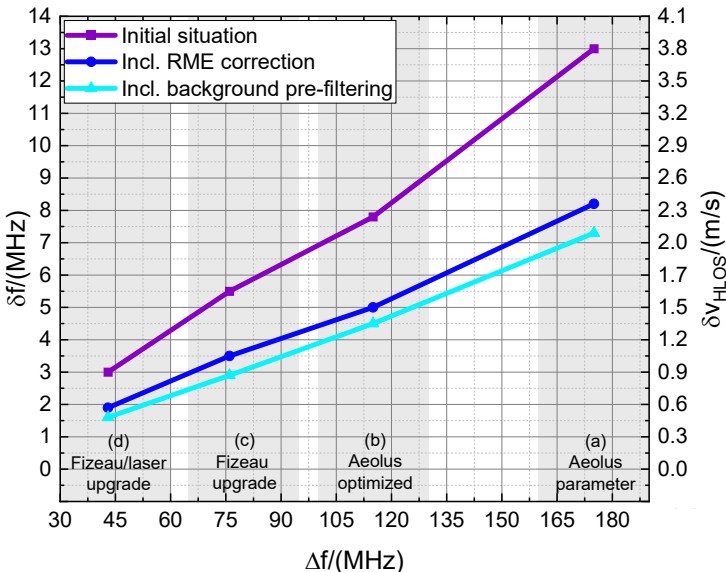

**Figure 17.** Wind measurement accuracy $\delta f$ (left y-axis) and $\delta_{v_{\mathrm{HLOS}}}$ (right y-axis), calculated for reasonable but low signal levels of $N_{\mathrm{S}} = 204.7$ ph.el. (140 LSB) and $N_{\mathrm{ped}} = 43.9$ ph.el. (30 LSB) for four sets (columns) of Fizeau instrumental parameters, calculated by means of Eqs. (9) and (13). $C_{\mathcal{V}} = 0.7$, $k_{\mathrm{r}} = 0.8$ and $r = 1.8$ is used for all cases. The initial situation is indicated by the purple line. Potential improvements for correction of the radiometric efficiency loss and optical pre-filtering are indicated by the dark-blue and light-blue line, respectively.

plane), as dictated by the Fizeau interferometer plate separation and wedge angle, together with appropriate plate reflectivity and fringe finesse. Two possible future systems are considered below.

**c) Reduce FSR and line width**

With approximate doubling of the interferometer plate spacing to $\approx 135\ \mathrm{mm}$ (instead of $68.5\ \mathrm{mm}$) and appropriate selection of wedge angle and finesse, the Mie signal collection efficiency and $N_{\mathrm{S}}$ may be maintained, and the overlap of orders as well as the value of $N_{\mathrm{ped}}$ per MHz is approximately doubled. A useful spectral range of about $\pm 400\ \mathrm{MHz}$ should readily be achievable to provide the required $v_{\mathrm{HLOS}}$ range of $\pm 100\ \mathrm{m\,s^{-1}}$. In this case, the Fizeau instrumental profile would have a FWHM of $\approx 47\ \mathrm{MHz}$ (Lorentzian shape) which, after convolution with the laser pulse profile of $\approx 45\ \mathrm{MHz}$ (Gaussian shape)

leads to a resultant fringe profile of $\Delta f = 76\ \mathrm{MHz}$.

Incorporation of these values in Eqs. (9) and (13) leads to $\mathrm{SNR}_{\mathrm{refined}} = 9.7$, $\delta f = 5.5\ \mathrm{MHz}$, and $\delta_{v_{\mathrm{HLOS}}} = 1.6\ \mathrm{m\,s^{-1}}$. With the RME factor, these become $\delta f = 3.5\ \mathrm{MHz}$ and $\delta_{v_{\mathrm{HLOS}}} = 1.0\ \mathrm{m\,s^{-1}}$, and with further optical pre-filtering, $\delta f = 2.9\ \mathrm{MHz}$ and $\delta_{v_{\mathrm{HLOS}}} = 0.8\ \mathrm{m\,s^{-1}}$. These 3 sets of values are indicated in column (c) of Fig. 17.

**d) Further reduce FSR, line width and laser pulse width**

Considered purely as a spectroscopic accuracy problem, there remain two powerful constraints on further advance. Firstly, the present laser pulse profile of width $\Gamma_{\mathcal{G}} = 45\ \mathrm{MHz}$ is now a major contributor to the fringe width $\Delta f$. Secondly, the meteoro-





logical requirement of $v_{\mathrm{HLOS}}$ ranging over $\pm100\,\mathrm{m\,s^{-1}}$, requires a large useful spectral range. It is thus worth examining the "what, if" question of reducing both.

Consider a laser pulse profile reduced to $\Gamma_{\mathcal{G}} = 20\,\mathrm{MHz}$, which would require an increase of the laser pulse length from about 20 ns to 50 ns. With further increase of plate separation to $\approx 200\,\mathrm{mm}$, a useful spectral range of about 280 MHz should be achievable, to provide a $v_{\mathrm{HLOS}}$ range of $\pm72\,\mathrm{m\,s^{-1}}$. With further appropriate selection of Fizeau parameters to maintain $N_{\mathrm{S}}$, the resultant fringe profile would be $\Delta f \approx 43\,\mathrm{MHz}$, with approximate trebling of the R-B background $N_{\mathrm{ped}}$ per MHz. Following the previous calculations, the accuracy values become $\mathrm{SNR}_{\mathrm{refined}} = 10.0$, $\delta f = 3.0\,\mathrm{MHz}$, and $\delta_{v_{\mathrm{HLOS}}} = 0.9\,\mathrm{m\,s^{-1}}$. With the RME factor, these become $\delta f = 1.9\,\mathrm{MHz}$ and $\delta_{v_{\mathrm{HLOS}}} = 0.6\,\mathrm{m\,s^{-1}}$. With further optical pre-filtering, these become 750 $\delta f = 1.6\,\mathrm{MHz}$ and $\delta_{v_{\mathrm{HLOS}}} = 0.5\,\mathrm{m\,s^{-1}}$. These 3 sets of values are indicated in column (d) of Fig. 17.

### 5.4.3 Comments and discussion of wind accuracy results

The most obvious feature of the accuracy values as represented in Fig. 17 is the rapid near-linear improvement with Fizeau fringe width $\Delta f$. In column (b), with operation at optimum AOI, $\delta_{v_{\mathrm{HLOS}}}$ has immediately reduced from $3.8\,\mathrm{m\,s^{-1}}$ to $2.3\,\mathrm{m\,s^{-1}}$, and to $1.5\,\mathrm{m\,s^{-1}}$ with further RME correction.

In column (c), $\delta_{v_{\mathrm{HLOS}}} = 1.6\,\mathrm{m\,s^{-1}}$, and with further RME correction would become $1.0\,\mathrm{m\,s^{-1}}$, which is a factor of 3.8 improvement on the starting value. These gains in wind measurement accuracy are, of course, significant. However, possibly of greater significance is the prospect of considerably increased global coverage. Simple analysis shows that, to first order, similar factors of improvement would be achieved starting from very much larger and presently 'not useful' values of $\delta_{v_{\mathrm{HLOS}}}$.

As obvious from Fig. 17, the final improvements in accuracy would be achieved with optical pre-filtering and reduced laser 760 pulse profile; it is worth considering their technical feasibility. Techniques of optical pre-filtering and FSR extension were extensively developed in the era of classical spectroscopy [for an early review see for instance Chapter 6 in Vaughan (1989)]. Investigation of possible techniques for the present Fizeau instrument would not be trivial, but should be relatively low-cost without major impact on overall optical layout. However, reduction of the laser pulse frequency width would clearly involve a major and costly long-term programme. The existing Aeolus laser design concept is at least 25 years old. The present short 765 temporal pulse length, about 20 ns, leading to a physical length of about 6 m and width about 50 MHz, is driven by laser engineering and the required high-peak power to provide high conversion efficiency in the frequency tripling process. Ideally in the future, optimisation of the balance of pulse length, conversion efficiency and total pulse energy would seem desirable, and in performance terms, potentially cost-effective.

It is finally worth mentioning that the accuracies noted above are comparable to what may be achieved with heterodyne 770 detection wind lidar as, for instance, shown by by Witschas et al. (2017, 2020, 2022a, 2023a). In Witschas et al. (2017) for instance, the standard deviation of the LOS wind speeds derived from an airborne heterodyne detection wind lidar is $\delta_{v_{\mathrm{LOS}}} = 0.2\,\mathrm{m\,s^{-1}}$.





## 6    Conclusions

In its four and a half years of operation following its launch in August 2018, ESA's Aeolus mission has provided an enormous
volume of global wind lidar data. Initially intended primarily as a three-year technical demonstrator, the mission quickly proved
that the volume and quality of its data could significantly enhance current numerical weather prediction. Besides the Rayleigh-
clear winds, that provided a much larger number of data, Mie-cloudy winds were shown to be of particular importance due to
their higher precision and their availability in the boundary layer region.

Since launch, the detailed spectroscopic operation of the Fizeau interferometer has been extensively investigated against a
background of technical observation, revealing two important findings. Firstly, the apparent angles of incidence of the scattered
return signal beam on the Fizeau interferometer were considerably larger than anticipated, reaching up to several hundred
$\mu$rad, as clearly evidenced in the FPI-based Rayleigh channel Witschas et al. (2022b). Secondly, the signal levels were lower
than expected for both the Rayleigh and the Mie channels. From the Mie channel, viable returns were almost entirely due to
strongly enhanced scattering from cloud top, thin diffuse cloud, volcanic aerosols, dust plumes and smoke from forest fires,
with virtually no viable signals from background aerosol.

The present investigation has thus concentrated on fundamental spectroscopic problems with particular emphasis on studying
the composition of the spectral line width from the Fizeau spectrometer and its impact on the SNR, fringe shift and wind
measurement accuracy at the quantum limit of performance, as expressed in Eq. (9). This analysis underscores the critical
importance of minimising spectral line width. In terms of measurement accuracy, a twofold reduction in line width is equivalent
to fourfold increase in signal energy.

The study of Fizeau fringes from Aeolus operation establishes a large Gaussian shape contribution of $\approx 130\,\mathrm{MHz}$ (FWHM)
to the instrumental profile, which consists of a Lorentzian $\approx 100\,\mathrm{MHz}$ (FWHM). Wave-optic analysis of fringe formation
provides a convincing explanation for this effect, attributing it to the input signal beam's large AOI of $\approx 300\,\mu$rad. This large
angle also likely contributes to the observed loss of radiometric signal collection efficiency due to beam clipping at the small
field stop aperture in the optical train. In contrast, operation closer to normal incidence greatly reduces the Gaussian compo-
nent and should minimise any signal loss. The implications of these findings are discussed for several scenarios, including
the optimisation of present Aeolus Fizeau parameters, an upgraded Fizeau system, and an enhanced laser source. Projected
improvements in wind speed measurement accuracy range from 1.8 to 7.2 times over the demonstrated Aeolus performance.

From a broad perspective, the present findings regarding the apparent misalignment of the signal beam, amounting to $\approx$
$300\,\mu$rad, warrant further comment and discussion. This value is indeed large in the context of standard interferometric and
spectroscopic practices. In a laboratory setting, where direct micro-adjustment is possible, misalignment errors of less than
$10\,\mu$rad are typically expected. For Aeolus, the large and progressive changes of AOI in both the Rayleigh and Mie channels
require analysis and explanation. It is also worth noting that if a misalignment of $\approx 300\,\mu$rad results in a signal loss of $\approx 60\%$,
crude estimates suggest that a misalignment of $\approx 450\,\mu$rad could lead to losses exceeding $\approx 90\%$, assuming the AOI is one of
several potential root-causes of the signal loss (though this remains unverified). For instance, The Pierre Auger Collaboration
et al. (2024) noted that laser-induced contamination and laser-induced damage is the most probable causes for the progressive





observed signal loss during the mission and suggest that clipping accounts for less than $10\%$ of the loss during the analyzed mission phase. The also mention that the initial loss mechanisms are still subject of ongoing studies.

In conclusion, for future comparable lidar missions, it is desirable to address several questions arising from the Aeolus experience. These questions may include: What are the primary sources of alignment error in Aeolus? Are they attributable to distortions in the relatively complex optical train. Would it be feasible and beneficial to incorporate active optical control and micro-alignment for the signal beams incident on the interferometers? Furthermore, could a robust space-qualified scheme be developed for this purpose?

It is also worth mentioning that the results and tools presented in this study might be useful for optimizing the specifications for the Mie channel of any Aeolus-like successor mission.

*Code availability.*

The code that is used for data analysis (LabVIEW vi) and for figure plotting (OriginLab) can be provided upon request.

*Data availability.*

The particular data sets used in this study can be provided upon request.





# Appendix A: Wave-optics model

As mentioned in Section 3, the wave-optic Fizeau model was originally developed to investigate the effect of circular defects arising from the MRF process applied to the Fizeau plates (Vaughan and Ridley, 2013; Vaughan and Ridley, 2016). The impact of these on fringe formation was initially modelled by using the ray optic approach, where the component of the transmitted field after the $p^\mathrm{th}$ passage through the Fizeau has a phase of (Born and Wolf, 1980)

$$\delta_p = \frac{4\pi}{\lambda} h \cos(\theta) \frac{\sin[(p-1)\alpha]}{\tan(\alpha)} \left\{ \cos[(p-1)\alpha] - \tan(\theta)\sin[(p-1)\alpha] \right\}. \tag{A1}$$

Here, $\lambda$ is the wavelength, $h$ is the plate separation, $\alpha$ is the wedge angle and $\theta$ is the angle of incidence, defined to be positive when the incoming radiation is tilted away from the apex of the wedge. The circular defects from MRF polishing were modelled by a simple sinusoidal variation in the surface of one of the plates:

$$h(x,y) = h_0 + h_g \cos\frac{2\pi r}{w_g}, \tag{A2}$$

where $r^2 = x^2 + y^2$. It can immediately be seen that there are some issues in applying Eq. (A1). The surface variations can be included via $h$ and $\alpha$, i.e. by treating $\alpha$ as a local effective wedge angle. However, this approach will not be correct when there is a large plate separation or when $\theta$ departs significantly from zero, as it assumes a ray that comes from a particular location on the plate always returns close to that same location. For this reason, it was decided to investigate a wave-optics approach, as described in Section A1. A comparison between results using Eq. (A1) and corresponding wave-optic calculations are given in Fig. A1 for the following parameters: $\lambda = 355$ nm, $h_0 = 68.5$ mm, $h_g = 2$ nm, $w_g = 1$ mm, $\alpha = 4.77\ \mu\mathrm{rad}$.



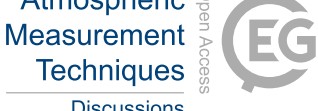

**Figure A1.** Four modelled fringe patterns in the presence of circular plate defects. Upper figures are ray-optic modelling, using two slightly different wavelengths to give a different fringe location in each figure. Lower figures are the corresponding wave-optics results.

It can be seen that the ray optics approximation gives a "zig-zag" pattern, whereas the correct structure has more "broken" appearance, with the fringe divided into separate segments. This broken structure was found to be in better agreement with experimental data for this type of plate defect.

In addition to better describing the effects of plate defects, the wave-optics approach allows for a natural treatment of the
range of incidence angles present in a lidar system by treating the input radiation as a Gaussian speckle pattern. This is described in detail in Section A3. Wave-optics also allows modelling of diffraction effects due to intensity variation in the input light,





e.g. arising from obscuration. Section A4 briefly outlines an alternative approach to including a range of incidence angles, still based on the wave-optic model, but using an incoherent sum of plane wave components.

**A1 Modelling approach**

The geometry of the Fizeau is sketched in Fig. 1, in the schematic of the Mie channel. Note that the wedge angle, $\alpha$, is greatly exaggerated: in the instrument under consideration, $\alpha$ is just $4.77~\mu\mathrm{rad}$ as listed in Table 1.

For a given input field $E_I$ we wish to calculate the output field $E_O$ and the output intensity $I_O = |E_O|^2$. Let $E_1$ be the field transmitted through the second plate after the light has undergone one pass of the Fizeau. This is given by

$$E_1 = t_A F\{E_I\} t_B, \tag{A3}$$

where $t_A$ and $t_B$ are the amplitude transmission coefficients of the plates $A$ and $B$, and $F$ is an operator that takes the field after the first plate and transforms it to the field before the second plate. For simplicity, the plates are taken to have zero thickness. The transmitted field after the light has made a further round-trip through the Fizeau is given by

$$E_2 = r_B r_A F^2\{E_1 \exp(2 i k \alpha x)\}. \tag{A4}$$

Here, it is the second plate, $B$, that is angled with respect to the $x$ direction, with plate $A$ parallel to $x$. Owing to the small 855 size of the angle, the effect of the wedge is simply a phase shift on the reflected radiation that is proportional to $\alpha$ and $x$. $k$ is the wave vector of the incident field, and $r_A$ and $r_B$ are the amplitude reflection coefficients of the plates. Since the reverse passage $B \rightarrow A$ is determined by the same operator as the passage $A \rightarrow B$, we use the notation $F^2$ for the double application of the operator $F$. Note that the transmission factor $t_B$ does not need to be applied again as it is already included in $E_1$ (see Eq. (A3)). Equation A4 can be extended to the $n+1^{\text{th}}$ pass through the Fizeau as follows:

$$E_{n+1} = r_B r_A F^2\{E_n \exp(2 i k \alpha x)\}. \tag{A5}$$

Finally, one can write,

$$E_O = \sum_{n=1}^{N} E_n. \tag{A6}$$

The value of $N$ is chosen to be sufficiently large that inclusion of further passages through the Fizeau has a negligible effect on the resulting fringe profile. The appropriate value of $N$ will depend on the plate reflectivities. For example, with both 865 plates having the same intensity reflection coefficients of $R = 0.88$, $N = 55$ is found to give satisfactory results. An order of magnitude estimate of the terms being neglected can be found by noting that, after $N$ round trips, the neglected fractional power inside the Fizeau is of order $R^{2N}$, which is $3.6 \times 10^{-5}$ in this case.





## A2 Solution of paraxial wave equation

The operator $F$ introduced in Section A1 is implemented by starting with the paraxial wave equation for monochromatic
radiation in Cartesian coordinates, which is

$$\frac{\partial^2 E}{\partial x^2} + \frac{\partial^2 E}{\partial y^2} - 2\,i\,k\frac{\partial E}{\partial z} = 0, \tag{A7}$$

where the wave is propagating in the $z$ direction, taken to be the direction normal to the first plate. Equation A7 is solved
using a Fourier-domain approach. Fourier transformation with respect to the $x$ and $y$ coordinates is applied to the equation.
The resulting ordinary differential equation in $z$ has the following solution

$$\overline{E}(\boldsymbol{\omega}, z_B) = \overline{E}(\boldsymbol{\omega}, z_A)\exp\left[\frac{i|\boldsymbol{\omega}|^2(z_B - z_A)}{2k}\right], \tag{A8}$$

Here, $\boldsymbol{\omega}$ is the spatial frequency vector, $\overline{E}(\boldsymbol{\omega}, z_A)$ is the Fourier transform of the known field in the $z_A$ plane and $\overline{E}(\boldsymbol{\omega}, z_B)$ is
the transform of the desired field in the $z_B$ plane. The field itself is produced via the inverse transform. The Fourier transforms
are calculated using the Fast Fourier Transform (FFT) algorithm, more detail can be found in Jakeman and Ridley (2006). The
input field is defined on a discrete grid. A square grid is used here, but a rectangular grid is also possible. Typically, the grid
consists of 1024 by 1024 points with a space step size of $50\,\mu\mathrm{rad}$. One potential issue with the FFT method is that it is periodic
in space, which can lead to edge effects. This is avoided by, first, making sure that the Fizeau fringe appears near the centre of
the grid, via precise setting of the wavelength, and second, tapering the input intensity to zero at the edges of the input wave.
This tapering uses a super-Gaussian profile of $10^{\mathrm{th}}$ order, i.e. $\exp\left(-|r/w_s|^{20}\right)$, where $r$ is a 2D position vector with origin at
the center of the grid, and the super-Gaussian radius is $w_s = 23.4\,\mathrm{mm}$. It is worth mentioning that the tapering does not effect
the region of interest which is the $36\,\mathrm{mm}$ Fizeau diameter. A cross-section through this tapered profile is shown in Fig. A2.

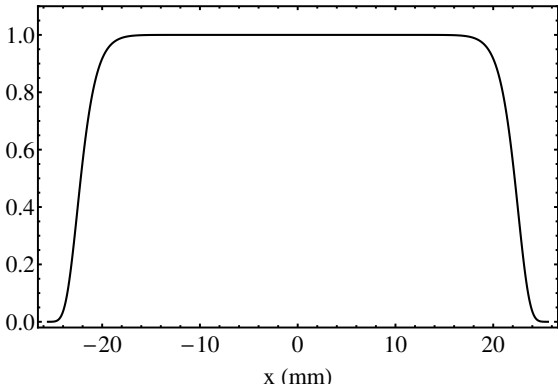

**Figure A2.** Tapered input profile.

It would also be possible to use a profile similar to that in Fig. A2 as an absorbing boundary applied to one of the mirrors.
However, it was found that tapering of the input wave amplitude worked well enough. Fig. A3 shows example Fizeau fringes





using three different values of: $N = 14$ (cyan), $N = 28$ (red), and $N = 55$ (black). The Fizeau parameters used for simulation are: a plate separation of $68.5$ mm, a wedge angle of $4.77$ $\mu$rad, reflectivity $R = 0.88$, and a wavelength of $355$ nm. The free

spectral range is $\lambda/2\alpha = 37$ mm and the length of the modelled region is $51.2$ mm. Thus, with the fringe at the centre of the region, the fringe corresponding to the adjacent FSR is well away from the edge, reducing the possibility of spurious diffraction at the edge contaminating the central fringe profile. Here, the incident field is a plane wave with a tilt angle of $300$ $\mu$rad relative to the $z$-axis (i.e. the normal of plate $A$). The convention used here is that $+ve$ angles tilt the input radiation in the direction of increasing plate separation (i.e. increasing because of the wedge).

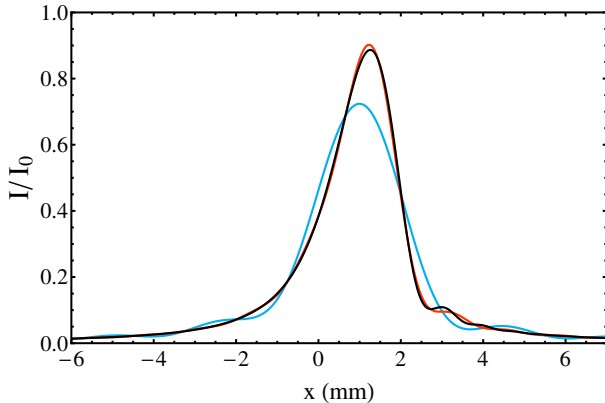

**Figure A3.** Fringe produced using increasing numbers of round trips for a tilt angle of $300\mu$rad. Cyan is $N = 14$, red is $N = 28$ and black is $N = 55$.

Here, the intensity is normalized by the input intensity. As mentioned previously, a value of $N = 55$ was considered sufficient for this work. A further indication of the errors involved in truncating the summation in Eq. (A4) is given in Fig. A4, which shows the difference in intensity of fringes produced with $N = 55$ and $N = 100$. It can be seen that the maximum difference is of order $2 \times 10^{-4}$, relative to the input intensity. Note that this is nearly an order of magnitude greater than the estimate in the previous section, based on total power. This discrepancy can be explained by the fact that the intensity differences are

both positive and negative, so there is significant cancellation when a difference in power is considered, because the power is a spatial integration of intensity.





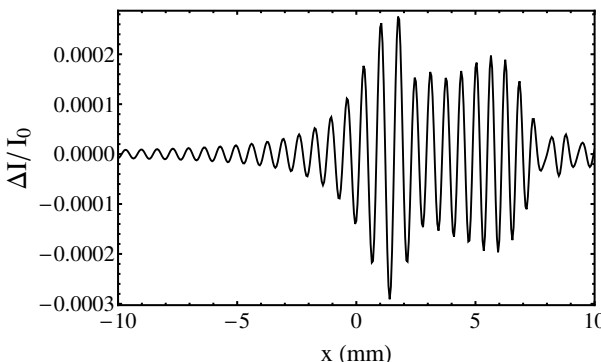

**Figure A4.** Relative differences in intensities for 55 loops versus 100 loops.

Note that, in this case, the ray-optics approach of Eq. (A1) gives an almost identical fringe profile, so this particular scenario of plane wave illumination and perfect plates does not necessitate a wave-optics approach.

**A3 Coherence and averaging**

In practice, in a pulsed Doppler wind lidar, the radiation incident on the Fizeau will not be a coherent plane wave. Therefore, the Fizeau model needs to take into account the actual structure of the input radiation. At any given moment in time, the laser illuminates a large number of scatterers within a volume determined by the spatial extent of the laser beam and its pulse duration. As a result, the field incident on the Fizeau interferometer will be a Gaussian speckle pattern. During the time corresponding to the range-gate of the system, the pulse will illuminate different sets of scatterers and the accumulated

fringe will be temporally averaged, which smooths out the speckle variations. In addition, there will be further smoothing when fringes from multiple pulses are summed. In the Aeolus spectrometer, spatial averaging also occurs because the detector readout is summed along the length of the fringe. It is worth reviewing and quantifying these various averaging/smoothing effects.

Consider a mono-static system, with the transmit/receive telescope producing a plane wave input to the Fizeau interferometer

from any individual (far-field) scatterer.

Consider, first, a single scattering layer situated at range $z = L$. For simplicity, we take $z = 0$ as being the position of the telescope entrance pupil. Assume the laser beam illuminating the scattering layer is a TEM$_{00}$ Gaussian, it has an intensity profile given by

$$I(r) = I_0 \exp\left(-2\frac{r^2}{w^2(z)}\right). \tag{A9}$$

The beam radius at the $1/e^2$ intensity point is

$$w^2(z) = w_0^2\left[1 + \left(\frac{2z}{k\,w_0^2},\right)^2\right], \tag{A10}$$





The backscattered light at the telescope pupil will form a speckle pattern. It can be shown that this speckle pattern will be characterized by a field correlation function of the form (Jakeman and Ridley, 2006)

$$\left| g^{(1)}(r) \right| = \exp\left[ -\frac{1}{2}\left( \frac{k\,w(L)\,r}{2\,L} \right)^2 \right], \tag{A11}$$

with $r$ being the distance between any two points in the $z = 0$ plane. Note that this is the modulus of the correlation function. The full, complex correlation function includes a phase term associated with the curvature of the wavefront returning from the scattering layer. However, because we have assumed a telescope focused on the scattering layer, this phase term is removed by the telescope and does not apply to the speckle pattern entering the Fizeau interferometer. Thus, in the numerical modelling of speckles, there is no phase term and the auto-correlation function is real rather than complex.

Employing the Siegert relation (Jakeman and Ridley, 2006), the intensity correlation function can be shown to be

$$g^{(2)}(r) = 1 + \exp\left[ -\left( \frac{k\,w(L)\,r}{2\,L} \right)^2 \right]. \tag{A12}$$

The speckle size can be characterised by a single correlation length, defined as the separation $r$ at which the exponential term in Eq. (A12) has reduced by $1/e$. When the scattering layer is in the far field, the correlation length is simply

$$r_{\frac{1}{e}} = \frac{\lambda}{\pi \theta_b}, \tag{A13}$$

where $w(L) = \theta_b L$. The beam divergence half angle $\theta_b$ is

$$\theta_b = \frac{\lambda}{\pi w_0}. \tag{A14}$$

In a mono-static Lidar, the system field of view is determined by the divergence of the illuminating beam and, in one dimension, is $2\theta_b$.

    We can define a correlation area as

$$A_c = \pi\, r_{\frac{1}{e}}^2. \tag{A15}$$

    If the telescope has the magnification $M$, then the angular factors at the input to the Fizeau interferometer will increase by $M$ and the length scales decrease by $M$. Thus, the field input to the interferometer is also a speckle pattern, with a correlation length reduced by a factor of $M$. The output of the interferometer will also be a speckle pattern but with more complicated correlation properties. Roughly speaking, it can be considered to be an interference fringe modulated by a random speckle

pattern. When averaged over many independent random speckle patterns, a smoothed fringe will emerge, the width of the fringes being influenced by the correlation length of the original speckle pattern (or, equivalently, the Lidar FOV): a larger FOV means a shorter correlation length (via Eq. (A13)), which will result in a broader fringe.

    Consider, first, the significance of the laser pulse duration. Fig. A5 shows a simplified situation where there are only two scattering layers at different distances from the instrument. Let the laser pulse duration be $\tau$, with the second scattering layer

separated from the first by a distance $\Delta z$, which is greater than $\tau c/2$. It is evident that the leading edge of the pulse reflected





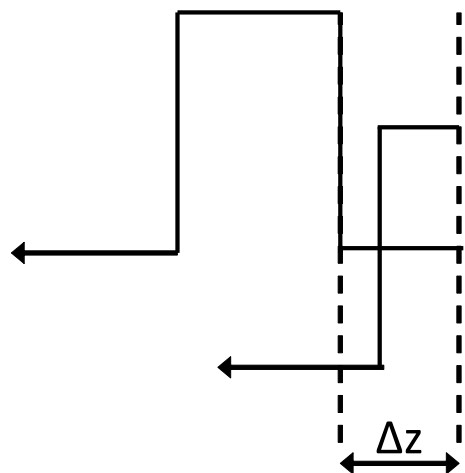

**Figure A5.** Backscatter from two separate layers.

from the second layer cannot overlap the trailing edge of the pulse reflected from the first. Thus, at the entrance to the spectrometer, there would be a certain speckle pattern for a period of time $\tau$ followed by a second, independent speckle pattern, also of duration $\tau$. In the more general case, where scatterers are uniformly distributed along the beam propagation path, the speckle pattern will vary continuously in time, with a characteristic time constant of $\tau$. Provided the time taken for light to

propagate through the interferometer is significantly shorter than the pulse duration, the same characteristic time constant $\tau$ applies to the output of the Fizeau interferometer. The temporal averaging is achieved simply by accumulating light on the detector for a time duration longer than the speckle correlation time. The number of averages can be taken to be the ratio of this detector time-gate to the speckle correlation time. In terms of the vertical spatial resolution of the wind measurement, $R_v$, the number of speckles averaged is simply

$$N_1 = \frac{2R_v}{\tau c}. \tag{A16}$$

Note that this assumes that the scattering cross-section is uniform along the length of the volume being probed by the laser beam. If the scattering cross-section is non-uniform, different scattering layers will reflect different quantities of laser light and the effective number of averages will be less than the result of Eq. (A16). In addition, Eq. (A16) assumes that the pulse is fully temporally coherent. If this is not the case, the pulse duration $\tau$ must be replaced by the (shorter) coherence time. The

spatial averaging is carried out after the speckle light has been detected: columns of pixels are summed along the direction of the fringe. The number of effective averages here depends on the size of the speckle and the size of the detector pixels. Using Eq. (A13) with $\theta_b = 6\,\mu\mathrm{rad}$ and $355\,\mathrm{nm}$ wavelength, gives a speckle size of $1.9\,\mathrm{cm}$ at the entrance to the telescope and thus a speckle area of $11.3\,\mathrm{cm}^2$. Dividing by the telescope magnification $M = 41.7$, gives a speckle size of $0.45\,\mathrm{mm}$ at the entrance to the interferometer. Eq. (A13) thus gives a speckle area at the entrance to the interferometer of $0.64\,\mathrm{mm}^2$. Assume

that there are the same number of speckles at the output of the interferometer and that the speckle size is the same across every column of pixels, i.e. it does not vary at different positions in the fringe pattern. Now, each detector pixel is a square





of $1.6\,\mathrm{mm}$ width and there are 16 pixels in a column. Thus, when the speckle pattern is averaged over a column there are approximately $N_2 = 41/0.64 = 64$ effective speckles averaged over. In fact, taking the effective number of speckles as the detector area divided by the speckle area is only exact in the limit of a large number of speckles. For a square detector area, the
exact result for the effective number of speckles is (Goodman, 2007)

$$N_2 = \left( \frac{1}{\sqrt{a}} Erf\left[\sqrt{\pi a} - 1\right] - \frac{1}{\pi a}\left[1 - \exp\left(-\pi a\right)\right] \right)^{-2}, \tag{A17}$$

where $a$ is the detector area divided by the speckle area. This formula gives $N_2 = 69.4$ as the effective number of speckles. Note, however, that this is still an approximation because the detector column is rectangular rather than square.

The third stage of averaging comes from accumulation of signal from multiple, independent pulses. If the number of pulses
averaged is $N_3$, we can define a total number of effective averages as

$$N_T = N_1\,N_2\,N_3. \tag{A18}$$

Now, the statistics of an individual speckle pattern are negative exponential (Jakeman and Ridley, 2006):

$$P(I) = \frac{1}{I_A}\exp\left(-\frac{I}{I_A}\right), \tag{A19}$$

where $I_A$ is the average intensity. Note that when we consider the output of the interferometer the average intensity is spatially
varying: high near the peak of a fringe and low in the troughs. The sum of $N$ speckle patterns follows a Gamma distribution (Jakeman and Ridley, 2006):

$$P(I) = \frac{I^{N-1}}{I_A^N \Gamma(N)}\exp\left(-\frac{I}{I_A}\right), \tag{A20}$$

where $\Gamma$ is the Euler Gamma function. This result can also be used when $N$ is not an integer. The average intensity is $N\,I_A$. The degree of fluctuation can be characterized by the second moment

$$\frac{\langle I^2 \rangle}{\langle I \rangle^2} = 1 + \frac{1}{N} \tag{A21}$$

Clearly, this takes on the value of 2 when there is no averaging and reduces to unity as $N \to \infty$. In the large $N$ limit, the detector output, in the absence of other noise sources, can be considered as a sum of a constant term and a smaller fluctuating term which varies from measurement to measurement. Thus, in this large $N$ limit, one can define an effective signal-to-noise ratio (SNR) as the constant term divided by the standard deviation of the fluctuating term. Using Eq. (A21), it is simple to
show that this SNR is just $\sqrt{N_T}$. As stated above, we have $N_2 = 69.4$. The minimum range gate is $2.1\,\mu s$ and the pulse duration 20 ns, which gives $N_1 = 110$. Thus, even without considering multiple pulse averaging, the minimum speckle SNR in normal operation is 87, which is sufficiently high to make this a small noise component compared to shot noise on the Rayleigh background. The number of laser pulses $P$ averaged on the CCD is $P-1$, and afterwards, $N \cdot (P-1)$ are averaged to 1 observation. $P$ has changed in the course of the mission and respective values are available in the ACCD paper by Lux
et al. (2024). The effect of speckle smoothing/averaging is included in the wave-optic model by calculating multiple fringes,





each starting with a statistically-independent speckle pattern and averaging the fringes to produce the final, smoothed, fringe. Typically, 100 different speckle patterns were used for each fringe. Note that it is not necessary to match the number of averages given by Eq. (A18): it is sufficient to use enough averages that residual speckle noise is negligible. Details of the generation of random speckle patterns can be found in Jakeman and Ridley (2006). The basis is the expression given by Eq. (A11) for the field correlation function. The Fourier transform of this is multiplied by an array of delta-correlated random complex Gaussian-distributed noise values. Inverse transformation yields two independent speckle patterns from the real and imaginary components. An example of a speckle-averaged fringe is shown in Fig. A6, compared with the plane wave fringe of Fig. A3. This uses the same central tilt angle of $300\ \mu\mathrm{rad}$ but with a Gaussian angular spectrum of $1/e$ width of $500\ \mu\mathrm{rad}$. Here, the

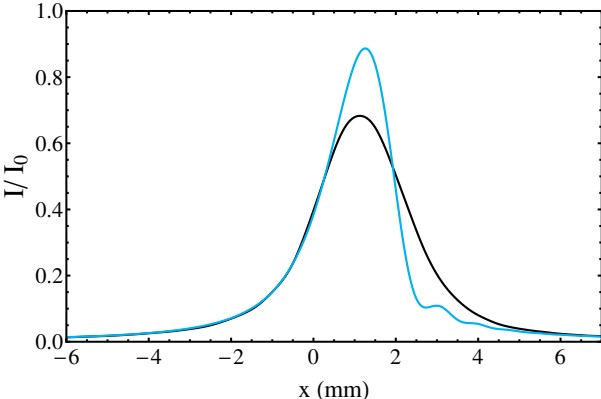

**Figure A6.** Fringe with a Gaussian angular spectrum (black) compared with a plane wave (cyan). The other parameters are the same as in Fig. A3.

two fringes have the same power, i.e. equal area under the fringe. Significant broadening of the fringe can be seen, as well as the influence of the Fizeau fringe asymmetry. It is worth noting that the broadened profile cannot be considered as simply a convolution of the plane-wave fringe with the angular spectrum because the shape of the plane-wave fringe is itself strongly dependent on the AOI.

This result used 100 speckle averages. An idea of the magnitude of the errors (i.e. the "speckle-noise") in this case can be found by calculating a second fringe with independent random speckles. The result of differencing the intensity of the two fringes is given in Fig. A7. Noting that one expects a square-root-of-two-increase in errors when differencing two random quantities, it can be inferred that the maximum intensity error is of order 2%.

## A4   Angular spectrum approach

The speckle approach described above treats the physics without any major approximations, but it does have the disadvantage of requiring averaging over many independent random speckle patterns to get an acceptably low error, which has implications for the time taken to compute a single fringe. If one is only interested in the incoherent fringe that results in the limit of infinite averaging, there is an alternative approach. This involves calculating many plane wave fringes for different angles of incidence





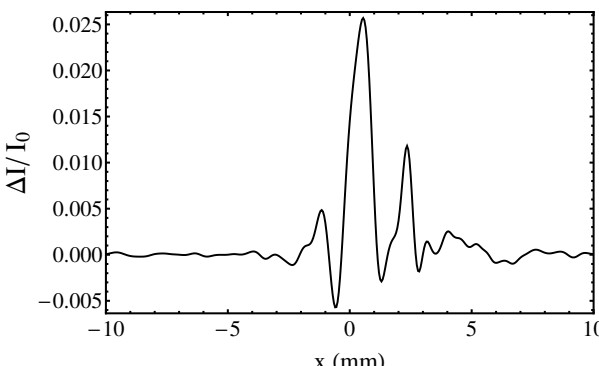

**Figure A7.** Residual errors from speckle fringe calculation from 2 independent simulations with $N = 100$ speckles.

and combining them by adding intensities rather than fields. A far field illuminating laser profile, such as given by Eq. (A9), can be converted into an angular spectrum, here denoted $\mathcal{A}$. This is the field angular spectrum, so the amplitude of $\mathcal{A}$ is the square root of the laser intensity at a given angle. If we consider the backscatter to come from a continuous scattering layer,
the field at the input to the Fizeau can be written as a two dimensional integral over the vector angle $\boldsymbol{\nu}$

$$\int \mathcal{A}(\boldsymbol{\nu}) \exp(i k \boldsymbol{\nu} r) d\boldsymbol{\nu}. \tag{A22}$$

The field at the output of the Fizeau can similarly be written as an integral over an angle-dependent transmission function $f$ (not to be confused with the earlier operator F as introduced in Eq. (A5)).

$$E_O = \int \mathcal{A}(\boldsymbol{\nu}) f(\boldsymbol{\nu}) d\boldsymbol{\nu}. \tag{A23}$$

Note that, here, the exponential phase factor in Eq. (A22) has been subsumed into $f$. Treating the scattering amplitude as a random variable (i.e. constant amplitude and random phase, the random phase arising from the random positioning of the scatterers), the incoherent fringe intensity can be written as an ensemble average, indicated by angled brackets in Eq. (A24).

$$I_O = \langle |E_O|^2 \rangle = \int \int \langle \mathcal{A}(\boldsymbol{\nu}) \mathcal{A}(\boldsymbol{\nu}') \rangle f(\boldsymbol{\nu}) f(\boldsymbol{\nu}') d\boldsymbol{\nu} d\boldsymbol{\nu}'. \tag{A24}$$

When the average is taken, only the radiation from individual scattering centres (e.g. single angles of arrival) adds in phase.
Cross-terms between different scattering centres are randomly phased and average to zero. This means the correlation function can be written as a delta function with respect to angle

$$\langle \mathcal{A}(\boldsymbol{\nu}) \mathcal{A}(\boldsymbol{\nu}') \rangle = |\mathcal{A}(\boldsymbol{\nu})|^2 \delta(\boldsymbol{\nu} - \boldsymbol{\nu}'). \tag{A25}$$

Substituting Eq. (A25) into Eq. (A24) results in the removal of one of the integrals, giving

$$I_O = \int |\mathcal{A}(\boldsymbol{\nu})|^2 |f(\boldsymbol{\nu})|^2 d\boldsymbol{\nu}. \tag{A26}$$

That is, the fringe profile is a weighted integral over a set of plane wave fringes. For numerical calculations, the integral is replaced by a sum over a suitably-chosen set of angles. In the results given here, the appropriate number of angles to use





was determined by comparing fringes with a different number of angles and also by comparing with the speckle approach. It was found that 81 angles in a 9 by 9 regularly spaced (square) array, with an angular spacing of 80 $\mu$rad, was acceptable for calculating a fringe using the parameters of Fig. A6. A cross-section through the centre of this array is shown in Fig. A8, where the solid line shows the Gaussian angular spectrum and the discrete points the angles used for each plane wave calculation. Figures A9 and A10 show differences between the 9 by 9 array and a result with 13 by 13 angles, and the 100 speckle average

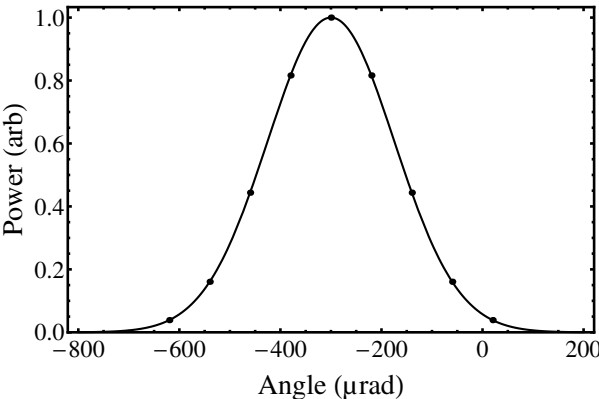

**Figure A8.** Angular spectrum approach to fringe calculation.

result of Fig. A6, respectively.

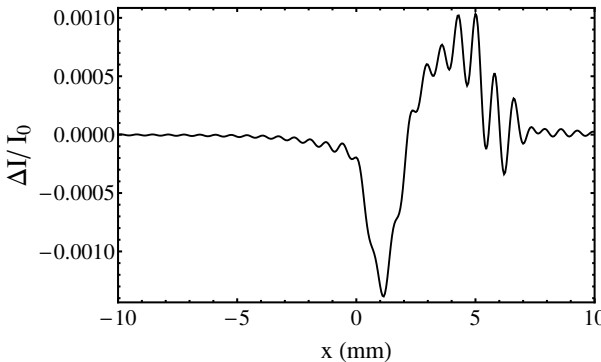

**Figure A9.** Differences between a fringe calculated with 9 by 9 angles and one with 13 by 13 angles.

These results imply that the angular spectrum is accurate with 81 angles. In fact, Fig. A10 suggests it is more accurate that the speckle result, with 100 speckles, because the differences are of similar magnitude to those seen in Fig. A7, meaning that errors in the speckle approach are the larger.

In terms of computation time, the angular spectrum approach is somewhat faster than the speckle approach in this case, because it uses 81 fringe calculations rather than 100, and because it does not require additional time to generate the random speckle patterns. However, the real computational advantage would come in cases where one needs to compute full two-





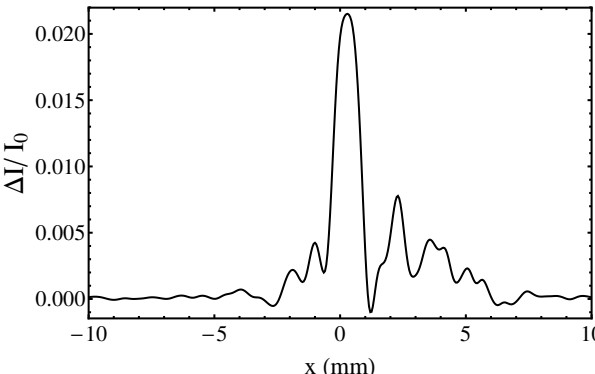

**Figure A10.** Differences between a fringe calculated with 9 by 9 angles and one using the speckle approach with 100 speckle averages.

dimensional fringes, without the averaging over detector columns. It was shown earlier in the discussion following Eq. (A16)

that this averaging is equivalent to $64$ speckle averages. So, if this additional averaging was not employed, $64$ times more independent speckle patterns would need to be used, with a proportional increase in computation time.





## Appendix B: Nomenclature

| | |
|---|---|
| $\Gamma_{\mathrm{FSR}}$ | Free spectral range |
| $\mathcal{P}_{\mathrm{Raw}}$ | Raw Signal fringe profile |
| $\mathcal{P}_{\mathrm{Las}}$ | Laser pulse profile |
| $\mathcal{P}_{\mathrm{Fiz}}$ | Fizeau Instrument profile |
| $\mathcal{P}_{\mathrm{Det}}$ | Detector channel spectral profile |
| $\mathcal{L}(x)$ | Lorentzian peak function |
| $\mathcal{I}_{\mathcal{L}}$ | Area under $\mathcal{L}(x)$ |
| $\Gamma_{\mathcal{L}}$ | Full width at half maximum of $\mathcal{L}(x)$ |
| $x_0$ | Center position |
| $\Gamma_{\mathrm{TH}}$ | Width of top-hat function |
| $H(x)$ | Heaviside step function |
| $\mathcal{L}_{\mathrm{px}}(x)$ | Pixelated Lorentzian profile |
| $\mathcal{I}_{\mathcal{L}_{\mathrm{px}}}$ | Area under $\mathcal{L}_{\mathrm{px}}(x)$ |
| $\mathcal{V}(x)$ | Voigt peak function |
| $\Gamma_{\mathcal{V}}$ | Full width at half maximum of $\mathcal{V}(x)$ |
| $\mathcal{G}(x)$ | Gaussian peak function |
| $\Gamma_{\mathcal{G}}$ | Full width at half maximum of $\mathcal{G}(x)$ |
| $\mathcal{I}_{\mathrm{fringe}}$ | Total fringe intensity |
| $\mathcal{I}_{\mathrm{peak}}$ | Fringe peak intensity |
| $L_{\mathrm{fraction}}$ | Fractional Lorentz contribution to $\mathcal{V}(x)$ |
| $G_{\mathrm{fraction}}$ | Fractional Gaussian contribution to $\mathcal{V}(x)$ |
| $p$ | Numerical value from Voigt tables |
| $L_{\mathrm{FWHM}}$ | Lorentzian FWHM contribution to $\Gamma_{\mathcal{V}}$ |
| $G_{\mathrm{FWHM}}$ | Gaussian FWHM contribution to $\Gamma_{\mathcal{V}}$ |
| $R$ | Reflectivity |
| $T$ | Transmission |
| $F$ | Finesse coefficient |
| $\varphi$ | Phase lag |
| $\mathcal{A}(\varphi)$ | The Airy function |
| $N_R$ | Reflectivity Finesse |
| $\Delta\tau$ | Laser pulse Duration |
| $\Delta\nu$ | Fourier-transform limit of laser spectral width |
| $\delta f$ | Standard Deviation of frequency estimates |





| | |
|---|---|
| $C$ | Spectral shape constant |
| $C_{\mathcal{L}}$ | Spectral shape constant for $\mathcal{L}(x)$ |
| $C_{\mathcal{G}}$ | Spectral shape constant for $\mathcal{G}(x)$ |
| $C_{\mathcal{V}}$ | Spectral shape constant for $\mathcal{V}(x)$ |
| $\langle N_S \rangle$ | Mean number of electron-counts |
| SNR | Signal to noise Ratio in shot noise limit |
| $\mathrm{SNR}_{\mathrm{basic}}$ | Basic SNR calculation with Background present |
| $m$ | Total number of detector Pixels |
| $\mathrm{SNR}_{\mathrm{refined}}$ | Refined SNR calculation with Background present |
| $N_{\mathrm{ped}}$ | Background signal per Pixel |
| $k_{\mathrm{r}}$ | Signal fraction within the analytical bandwidth |
| $f_{\mathrm{AB}}$ | Analytical bandwidth |
| $n$ | Number of pixels covered by $f_{\mathrm{AB}}$ |
| $r$ | Ratio of $f_{\mathrm{AB}}$ and $\delta f$ |
| $\delta_{v_{\mathrm{HLOS}}}$ | Standard deviation of the HLOS wind |

*Author contributions.* Michael Vaughan led the presented study, which started already back in 2013, and performed the main part of the presented analysis. Kevin Ridley developed the wave-optic model for the Fizeau spectrometer and performed all the corresponding simulations.
Benjamin Witschas performed the numerical simulations and supported the preparation of the manuscript. Oliver Lux provided the data of the Aeolus Fizeau fringes and support the corresponding analysis. Ines Nikolaus generated the Aeolus prototype fringes and supported the corresponding analysis. Oliver Reitebuch supported the data analysis and the preparation of the paper manuscript.

*Competing interests.* The contact author has declared that none of the authors has any competing interests

*Acknowledgements.* The authors gratefully acknowledge that initial studies of wave optic modelling applied to Fizeau interferometry were supported by ESA at Optical & Lidar Associates with two purchase orders (5401001330 and 5401002470). Part of this work was performed in the framework of the Aeolus DISC Phase E2 and Phase F activities also funded by ESA (40000126336/18/I-BG, 4000144330/24/I-AG). With the greatest respect and appreciation, the authors offer tribute to the memory of Alain Culoma (ESA), for his farsighted and enthusiastic support for the development of wave-optic modelling techniques. The authors also would like to thank Eric Jakeman for helpful discussions
during the development of the wave-optic model and Christian Lemmerz for his contributions during the investigations discussed in this paper.





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
