# Peer review of "Spectral performance analysis of the Fizeau interferometer onboard ESA's Aeolus wind lidar satellite"

_Atmospheric Measurement Techniques, 2024_

## Author Comment (AC1)

**(Author response)**

**Referee #1**
This paper, titled 'Spectral performance analysis of the Fizeau interferometer onboard ESA's Aeolus wind lidar satellite', investigates the factors influencing the measurement accuracy of wind speed by the Fizeau interferometer. The Fizeau interferometer serves as a critical component in the wind lidar system. In this study, the instrument function of the line shape is analyzed based on the Fizeau fringes observed from Aeolus. Additionally, the broadening effects due to angle of incidence (AOI), field of view (FOV), laser pulse characteristics, and defects are numerically examined. Furthermore, the impact of line broadening on fringe shift and Doppler wind measurement accuracy, as well as the influence of Rayleigh-Brillouin background signals on signal-to-noise ratio (SNR) and measurement accuracy, are discussed. Finally, the potential performance of upgraded Fizeau systems with optimized parameters is proposed for future applications. This work is highly significant and merits publication. I recommend accepting after minor revisions.
Thanks a lot for performing the review of our paper manuscript and for suggesting this work to being published in AMT after minor revision. We will answer to each of your comments below (blue color). The corresponding changes in the manuscript are highlighted in green color.

1.  In the analysis of contributory factors to the Fizeau fringe profile, the authors identify several key elements that influence the profile, including nonlinear fitting procedures, laser pulse characteristics, field of view (FOV), angle of incidence (AOI), plate defects, fringe skewness, and the impact of Rayleigh-Brillouin scattering. Could the authors provide a detailed analysis of how each factor affects the accuracy of Fizeau fringe characterization, particularly in terms of full width at half maximum (FWHM), and consequently, the precision of the final wind measurement?
    Thank you very much for your thoughtful comment. As you mentioned, we put significant effort into characterizing and quantifying the contributors to the Fizeau fringe profile. In Section 3, we examine and analyze the measured fringe profiles, revealing a large Gaussian contribution that results in an overall Voigt-shaped fringe. In Section 4, we use wave-optic model analyses and numerical simulations to investigate individual contributors, identifying two primary factors that determine the width and spectral shape of the Fizeau fringe. These are (1) the Gaussian shape of the laser pulse and (2) the interaction at the Fizeau, where the large AOI and FOV contribute significant Lorentzian and Gaussian spectral components. Since this information was previously spread across different subsections, we have added a dedicated subsection (4.6) for further clarification: 4.6 Summary of line broadening factors: From the extensive analysis of actual Aeolus fringe profiles in section 3, and the modelling and simulation studies of the present section 4, it is evident that two primary factors determine the width and spectral shape of the Fizeau fringe. These are firstly the Gaussian shape of the laser pulse, and secondly the interaction at the Fizeau of the large AOI and FOV that contribute large Lorentzian and Gaussian spectral components. The following section 5 investigates, at a fundamental theoretical level and from the evaluation of characteristic Aeolus fringes, how the actual width and spectral shape affects the measurement accuracy.

2. In this paper, the uniformity of the laser spot is not mentioned, does it affect the final profile of Fizeau fringe?
Thanks a lot for raising this point. It is true that the effect of the uniformity of the laser spot and its impact to the fringe width was not clearly mentioned. For further clarification regarding that topic, we added an explanation to the paper following line 411: Here, we are assuming a Gaussian-profiled FOV, which corresponds to an illuminating beam with a $TEM_{00}$ Gaussian profile. This is a smooth, uniform laser spot which neglects any fine scale structure that might exist on the beam, caused by the telescope obscuration, for example. However, any fine scale structure, if present, would not change the width of the Fizeau fringe because it would be smoothed out by the Fizeau response function. What would have an impact on the fringe width would be a wider than expected laser spot, or one with side-lobes outside of the central spot. We note, however, that light backscattered from side-lobes would be blocked by the field-stop and not lead to an increase in fringe width.

3. I think the program of the ALADIN has been demonstrated, the author analyzes the instrument function and showed the AOI a few hundred μrad, so could you give an explanation what kind of changes make a so large AOI?
Thank you very much for this valuable comment. Indeed, the analysis presented in this paper, as well as earlier studies based on data from the Aeolus Fabry-Perot interferometers (Witschas et al., 2022), revealed that the receiver was not perfectly aligned, with deviations of a few hundred μrad. As shown in the receiver sketch in Fig. 1, the Aeolus receiver follows a complex design in which the spectrometers are arranged sequentially to utilize the reflected light and thus "recycle" photons. Naturally, such a complex arrangement poses challenges in alignment, and any uncertainty in alignment may propagate through the system. Therefore, it is beyond the scope of this paper to determine which specific optical component was responsible for the misalignment. However, our analysis confirms its existence and highlights the need for critical on-ground verification in future systems. It is worth noting that it is not yet clear if the misalignment was already present on ground or occurred during launch of in space.

4. The author shows the influence of the defects such as sinusoidal and surface defects on the designed Fizeau, does the spherical defect of the mirror broaden the fringes?
The wave optic simulation results presented in Fig. 11 include the effects of all departures from surface flatness. We drew attention to the effect of the local wedge (i.e. linear) defect (in line 467) because it made a notable difference to the fringe width. Other defects, such as a spherical component, are undoubtedly present but don't have as much of an effect on the width. We didn't include a 2D surface map, which would give further insight into the various deviations from surface flatness, for reasons of space.

5. On line 405, Fig. 6 b should be Fig. 6 (b); line 555, the Fig.14 a should be Fig.14(a) and the same as shown online 574, please have a check.
Thanks a lot for pointing to these typos, which we have corrected accordingly in the revised version of the paper manuscript.

6. On line 650 for LSB an explanation should be made.
Thanks for the hint. We added an additional explanation. The paragraph now reads: "First, the mean value of $N_S$ is of the order of 1000 LSB, which corresponds to ≈ 1462 ph.el.. It is worth noting that during the detection process, the amplified detector signal including DCO is converted into units of least significant bits (LSB), and the conversion rate of this process is given by the radiometric gain of 0.684 LSB/ph.el. for the Mie ACCD detector (Lux et al., 2024). Hence, the fringes…".

7. The unit of AOI in line 434 has a writing error.
Thanks a lot. The Typo was corrected accordingly (MHz was changed to μrad)

---

## Author Comment (AC2)

**(Author response)**

**Referee #2**
This is an excellent paper. I found it a pleasure to read and review. It describes a detailed analysis of the Fizeau interferometer used to estimate Mie-scattered winds as part of the Aeolus mission. The paper is well-organized, describing the Aeolus mission and the role of the Fizeau interferometer, the interferometer itself, and then logically proceeding to characterization of the interferometer on the ground and a summary of the performance of the component during the mission. It concludes with a very credible hypothesis for the observed reduced performance observed on-orbit, and describes improvements that could be incorporated into the interferometer design for a follow-on mission that could significantly improve performance.
The paper is both informative and tutorial. I guess I was a bit surprised that a Fizeau interferometer has not previously been analyzed using a wave-optics approach, but the utility of the approach for this analysis is certainly justified and appropriate.
In my opinion, this paper could be published without revision. If the paper is returned to the authors, a couple of descriptions in the text could benefit from a bit more explanation.
Thanks a lot for performing the review of our paper manuscript and for suggesting this work to being published in AMT. It is great that both the informative but also the tutorial character of the paper manuscript is acknowledged, as it was not trivial to find a balance between both while writing. Thank you. We will answer to each of your comments below (blue color). The corresponding changes in the manuscript are highlighted in green color.

- Line 546: The sentence beginning with "The simulation analysis…" notes that the frequency estimation algorithm was modified. A bit more discussion here on the frequency estimation algorithm would be informative.
  It seems that this phrase was misleading, as the only modification was the addition of the background as a free fit parameter. However, we recognize that we did not explain the fitting routines in sufficient detail. Therefore, we have added the following sentences to Section 5.1.2, where the fitting is first used (following line 525):
  For the analysis of the simulated fringes, a non-linear square fit of Eq. (2) was applied, using the center position $x_0$, the FWHM $\Gamma_L$ and the area under the peak $I_L$ as free fit parameters.
  In addition, in section 5.1.3., we add the following side note (following line 547):
  … was modified to account for a background pedestal of unknown height, adding and offset term to Eq. (2) as a free fit parameter.

- Line 640: Although the values for r and fab agree with those of specified in the previous section, it isn't clear to me how they were determined. Were these determined through simulations?
  We agree that both the explanation of the Aeolus parameter used and the derivation of r and fab were not provided in sufficient detail. This was partly due to the need to shorten the manuscript to address length constraints. To ensure the necessary information is included, we have added the

following paragraph at line 635: Considering the earlier discussions in Sect. 3 and Sect. 4, as well as an examination of the Aeolus Fizeau Mie profiles, it is considered that (before detection) these profiles are close to a Voigt profile with a FWHM of approximately 175 MHz. This profile is further described as consisting of an instrumental Lorentzian component of ~95 MHz, as retrieved from the simulations shown in Fig. 8 b, and a Gaussian component of ~117 MHz, the latter resulting from the combination of 108 MHz (instrumental AOI aperture broadening) and 45 MHz (laser pulse width). Notably, when this Voigt profile is convolved with the detector's 'top-hat' function of 100 MHz pixel width, the resulting prototype fringe has an FWHM of 205 MHz, which is very close to the one shown in Fig. 2b. From these values, the Lorentzian fraction is estimated as L = 95/175 ≈ 0.54, with a corresponding $C_v ≈ 0.66$, resulting from numerical simulations similar to the one shown in Fig. 14. Using these values and following Eq. (13), extensive simulations of the SNR versus r reveal a weak, broad peak (not shown) which results in an optimal analytic bandwidth of 300 MHz. The derivation of these parameters is furthermore illustrated in the figure shown below which is not included to the manuscript to address length constraints. The left panel shows modeled Voigt profiles (FWHM = 175 MHz, $L_{fraction}$ = 0.54). The colored area indicates the area included by respective analytic bandwidths, in particular $f_{AB}$ = 175 MHz (top), $f_{AB}$ = 245 MHz (middle) and $f_{AB}$ = 455 MHz (bottom). The right panel shows the evolution of $k_r$ depending on $L_{fraction}$ for different $f_{AB}/r$ ratios (left y-axis). The black line indicates the respective $C_{CR}$-value (right y-axis). The values resulting for the Aeolus parameters is depicted by the orange dashed line. This is just to illustrate that detailed numerical simulations have been performed to end up with the values given in the manuscript.